

# Modeling precipitation $\delta^{18}$O pariability in East Asia since the Last Glacial Maximum: temperature and amount effects across different time scales

Xinyu Wen[1], Zhengyu Liu[2], Zhongxiao Chen[3], Esther Brady[4], David Noone[5], Qingzhao Zhu[1], and Jian Guan[1]

[1]Laboratory for Climate, Ocean and Atmosphere Studies, Dept. of Atmospheric and Oceanic Sciences, School of Physics, Peking University, Beijing, China
[2]The Center for Climatic Research, The Nelson Institute for Environmental Studies, University of Wisconsin-Madison, Madison, WI, USA
[3]Nanjing University of Information Science and Technology, Nanjing, China
[4]Climate and Global Dynamics, Earth System Laboratory, NCAR, Boulder, CO, USA
[5]Cooperative Institute for Research in Environmental Sciences, University of Colorado, Boulder, CO, USA

*Correspondence to:* X. Wen (xwen@pku.edu.cn)

**Abstract.** Water isotope in precipitation has played a key role in the reconstruction of past climate on millennial and longer timescales. However, for mid-latitude regions like East Asia with complex terrain, the reliability behind the basic assumptions of the temperature effect and amount effect are based on modern observational data and still remains unclear for past climate. In the present work, we re-examine the two basic effects on seasonal, interannual, and millennial timescales in a set of time
slice experiments for the period 22ka thru 00ka using an isotope-enable AGCM. Our study confirms the robustness of the temperature and amount effects on the seasonal cycle over China, with the temperature effect dominating in northern China, and the amount effect dominating in deep southern China, but no one distinct in the transition region of central China. However, our analysis does not show significant temperature and amount effects over China on millennial and interannual timescales, which is a challenge to those classic assumptions in past climate reconstruction. Our work helps shed light on the interpretation
of the proxy record of $\delta^{18}$O from modeling point of view.

## 1 Introduction

Stable water isotopes have been recognized as key tracers for tracking temperature footprint and moisture sources in air masses (Sturm et al., 2010; Noone, 2008). Based on the present relationship between $\delta^{18}$O records and temperature/precipitation, ear-lier observational studies suggest that the $\delta^{18}$O-temperature relationship at high latitudes tends to be associated with the "local
temperature effect", whereas the $\delta^{18}$O-precipitation relationship in the tropics and low latitudes tends to be associated with the "amount" effect (Dansgaard, 1964). Thus, the $\delta^{18}$O records in high-latitudes ice cores have long been used to infer temperature variability in past climate through the empirical temperature effect assumption (Schneider and Noone, 2007; Schotterer and Oldfield, 1996). Recently, there are explosive studies of oxygen isotope records from speleothems (Wang et al., 2001, 2005, 2008; Cheng et al., 2009; Chu et al., 2012), ice cores (Davis and Thompson, 2004; Davis et al., 2005; Yuan et al., 2004;



Thompson et al., 2000, 2006), tree rings (Feng et al., 1999; Grießinger et al., 2011), and lake sediments (Morrill et al., 2003, 2006; Zhang et al., 2011) found in the monsoon region, especially in the East Asia. These studies tend to suggest these isotope records reflect the East Asia Summer Monsoon (EASM) rainfall through the linkage of amount effect. This interpretation of monsoon intensity, however, has been challenged recently by isotope modeling studies, which suggest potentially significant

deviations from these presumed amount effects (LeGrande et al., 2006; LeGrande and Schmidt, 2009; Pausata et al., 2011; Johnson, 2011; Liu et al., 2014b). One basic and key question is that, is the present day relationship between $\delta^{18}$O and temperature/precipitation in East Asia for, say, seasonal cycle, also valid for interpreting past climate at millennial or longer time scales?

East Asia locates at the transition zone between temperature effect dominant high-latitudes and amount effect dominant low-

latitudes and thus its climate is influenced by both. As a result, the interpretation of $\delta^{18}$O in precipitation still remains as a great controversy, especially on diverse timescales. Modeling $\delta^{18}$O using isotope-enable GCMs provides a unique and quantitative approach to re-examine these basic relationships with temporal continuous data and complete spatial coverage (Vuille et al., 2005; Lee et al., 2007, 2008). The simulated temporal slope or correlation between $\delta^{18}$O and surface temperature/precipitation at various regions around East Asia might confirm or challenge those traditional linkages: Is it an appropriate way to interpret

$\delta^{18}$O record as the direct consequences of local temperature/precipitation changes on all the timescales?

Here, we will examine the temperature effect and amount effect over China using an isotope-enable atmosphere general circulation model (GCM) (Noone, 2008). The results show that the robustness of $\delta^{18}$O-climate relations highly depends on timescales. The classic relations derived from seasonal cycle timescale might be challenged on interannual and millennial timescales. It is therefore suggested that one should be cautious when applying the relations of two effects in reconstructing

paleoclimate using $\delta^{18}$O proxies. The paper is organized as follows: the model, experiments, and the observations for comparison are described in section 2; the model results on seasonal, interannual, and millennial timescales are discussed in section 3. The conclusion remarks are summarized in section 4.

## 2   Model and data

We will study the water isotope response using the isotope-enabled isoCAM3, which incorporated stable water isotopes into

the NCAR CAM3 (T31) (Noone, 2008). We performed time-slice experiments for the last 22,000 years with each succeeding snapshot  1,000 years apart, i.e., 22ka, 21ka, 20ka... 2ka, 1ka, and 0ka. These experiments are forced by the realistic green house gases (GHGs) concentrations, orbital parameters, land ice sheet and land-ocean mask, as well as the monthly sea surface temperature (SST) and sea ice fraction from a transient simulation of the last 22,000 years simulated in CCSM3 (Liu et al., 2009, 2012, 2014a). In addition, the $\delta^{18}$O values at sea surface, which is referred to as Standard Mean Ocean Water (SMOW)

at present, was linearly prescribed from 1.6‰ (22ka) to 0.5‰ (0ka) accounting for its change arisen from the fluctuation of sea level during the deglaciation. More details of isoCAM3 can be found in Noone and Sturm (2010). Each slice was integrated for 50 years and the last 40-year model outputs will be used in the analysis.



Global Network of Isotopes in Precipitation (GNIP) data will be used for model-observation comparison for the present. We collected monthly mean $\delta^{18}$O in precipitation from 31 stations over China covering the period 1961-2005. This dataset has sufficient spatial coverage. But majority of the records concentrate on two periods: 1986-1993 and 1996-2003, as shown in Fig 1(b). Moreover, there is only 12 stations with more than one year having continuous 12-month observations, showing as black

boxes in Fig 1(b). Thus, the length and discontinuity of the data apparently confine its usefulness in interannual-to-decadal timescale analysis. We average all the available records on each month for each station to derive the mean seasonal cycle of $\delta^{18}$O in precipitation for the following comparison.

## 3    Results

### 3.1    Seasonal cycle

We first examine the water isotope variability of the modern climate. The seasonal cycle of $\delta^{18}$O in precipitation in China is significantly affected by East Asia's monsoon climate, the East Asian summer monsoon (EASM) in June-July-August and the East Asian winter monsoon (EAWM) in December-January-February. Thus, the seasonal cycle of $\delta^{18}$O is dominated by the southwesterly moisture transport from the South Asia and Indian Ocean as well as the western North Pacific Ocean in summer, but by the westerly and northerly winds from high-latitude cold air mass in winter.

We first compare the seasonal cycles of the precipitation $\delta^{18}$O in the 00ka snapshot simulation with the GNIP observations in China, to examine the model's ability in reproducing the key variable that might be linked with both temperature and amount effects. The seasonal cycle of monthly mean $\delta^{18}$O in precipitation, surface temperature, and total precipitation are compared for eight regions in China (Fig 2, see Fig 1c for the definition of the regions). For each region, the modeled seasonal cycle are derived from the last 20years of the 00ka time slice (Fig 2, right column), whereas the observed seasonal cycle are selected

from the GNIP station that has the longest records in that region (Fig 2, left column). The correlation coefficients for $\delta^{18}$O with temperature and precipitation are marked on the top-right corner of each subplot.

At all sites in China, the temperature and precipitation exhibits a simple seasonal cycle, with a summer maximum and winter minimum in the observations, which is typical in the East Asian monsoon region (Fig2, left column). These seasonal cycles are largely reproduced in the model in the different regions across China (Fig.2, right column). The seasonal cycle of $\delta^{18}$O, and its

covariance with temperature and precipitation, however, is more complex, and can be grouped into three typical and distinct "modes".

The first mode is the "northern mode", which is dominant in the observations in northeastern China (Fig 2a), northern China (Fig2c), and northwestern China (Fig2e). This mode is characterized by a single summer maximum/winter minimum in $\delta^{18}$O that correlates positively with the temperature and precipitation. This seasonal cycle of $\delta^{18}$O is largely reproduced in

the model (Fig2b, d, f). We note that Tibet (Fig2o, p) should also be included in this mode, although the observation shows irregular variations of $\delta^{18}$O, most likely due to the short period of the record, especially in autumn and winter. The $\delta^{18}$O in the northern mode is dominated by the temperature effect, as in those cold high-latitude regions, such as Greenland and Antarctica





(Lee et al., 2007). Indeed, the annual mean temperature of these regions is the coldest in China, ranging from -5ºC in NE China to +4ºC in NW China.

The second mode is the "southern mode", which is dominant in the South China Sea (SCS) and the surrounding southern cities, such as Hong-Kong and Hai-Kou (Fig2k, m). Opposite to the northern mode, the $\delta^{18}$O of the southern mode exhibits

a single summer minimum/winter maximum in $\delta^{18}$O, which correlates negatively with the temperature and precipitation. The $\delta^{18}$O evolution is clearly dominated by the amount effect, with a greater rainfall amount leading to more depleted $\delta^{18}$O. The model is able to reproduce this $\delta^{18}$Oevolution in the SCS (Fig 2n). In southern China, the model is able to simulate the summer $\delta^{18}$O minimum, but not the winter maximum (Fig 2l). Instead, the modeled $\delta^{18}$O in southern China exhibits a double maximum in spring and fall, which resembles the third mode to be discussed next.

The third mode is the "central mode", which is dominant in the observation in central China (Fig 2g), southwest China (Fig 2i), and southern China (Fig 2k). The central mode $\delta^{18}$O is characterized by double maxima in spring and autumn sandwiched by a strong summer minimum. In this mode, the correlation between $\delta^{18}$O and temperature/precipitation tends to be insignificant. For example, central China is a typical central mode region (Fig2g), showing correlation coefficients as low as -0.1 between $\delta^{18}$O and temperature/precipitation. The model is able to simulate this double peak structure of $\delta^{18}$O in central

China (Fig 2h), southwestern China (Fig 2j) and southern China (Fig 2l). This more complex behavior of the $\delta^{18}$O seems to reflect the nature of the complex climate in these regions, which is located in a transition area between the northern mode, where the temperature effect is dominant, and the southern mode, where the amount effect tends to dominate. Therefore, the interpretation of the $\delta^{18}$O is more complex and should be cautious. This has important implications to the interpretation of the stalagmite proxies, which are found mostly in this area, such as the Hulu cave $\delta^{18}$O records near Nanjing city (Wang et al.,

2001). Thus, we would suggest that one should NOT interpret the $\delta^{18}$O records around this region simply as the monsoon rainfall amount.

In spite of the success of isoCAM3 in capturing the phase of the seasonal cycle of $\delta^{18}$O over most regions in China, the model has some deficiencies in quantitatively reproducing the annual range of $\delta^{18}$O. Compared with the GNIP observation, the seasonal cycle of the model $\delta^{18}$O is 3‰ (about 25%) lower for "northern mode" region, 0.5‰ (less than 10%) higher

for "southern mode" region, and is about comparable for "central mode" region with the observation. Overall, the model-observation comparison suggests that the model is capable in reproducing the major features of the seasonal climatology of $\delta^{18}$O in different regions of China, which seems to be controlled by complex processes of both temperature and amount effect in different regions.

The temperature effect dominates the $\delta^{18}$O-temperature relation over most of the northern China, while the amount effect

dominates the $\delta^{18}$O-precipitation relation over the southern China and SCS; the central transition region, however, seems to be controlled by a mixture of mechanisms. This distinctively different three regions can be seen in the correlation map between the seasonal cycles of $\delta^{18}$O and temperature (Fig 3a) and precipitation (Fig 3b) in the model, which shows a positive correlation in the north, negative correlation in the south, and leaving a blank area in the central transition latitudes of China. Thus, in the following discussions, we use "temperature effect" when referring to positive $\delta^{18}$O-temperature correlations, and "amount

effect" when referring to negative $\delta^{18}$O-precipitation correlations.



## 3.2 Interannual variability

In sharp contrast to the seasonal cycle, there is little correlation between the interannual variability of $\delta^{18}$O and temperature/precipitation. This can be seen in the point-by-point correlation coefficients between annual mean $\delta^{18}$O (weighted by monthly mean precipitation) and annual mean surface temperature (Fig 3c) and precipitation (Fig 3d) using the last 40 years

of model output of the 0ka time slice. The area with significance level greater than 90% is shaded in colors, we can see in Fig 3c and d that neither temperature effect nor amount effect is significant over most of China. In the mean time, there is a slight correlation of temperature effect over central China and amount effect over South China and SCS. Therefore, in this model, in spite of a seemingly significant implication of temperature and precipitation effect in the seasonal cycle over different regions in China, the interannual variability of water isotope, in general, is not a good representation of temperature or rainfall

variability. This seems to be consistent with a previous analysis (Maher, 2008; Dayem et al., 2010). In comparison, however, the amount effect seems to be significant over most of the South Asia monsoon region. This different role of amount effect between South Asia and East Asia monsoon regions reflect likely the different nature of monsoon moisture source between the two monsoon regions: the former derives its moisture source mostly locally from the nearby Indian Ocean, while the latter derives the moisture remotely from the Indian Ocean and western North Pacific (Pausata et al., 2011; Liu et al., 2014b).

## 3.3 Millennial variability

We now further show that the change of the model $\delta^{18}$O climatology at millennial timescale does not reflect the change of temperature and precipitation in China either. This can be seen in the cross-snapshot correlation coefficients between the millennial climatological variability of the annual mean $\delta^{18}$O (weighted with precipitation) and surface temperature (Fig 3e) as well as precipitation (Fig 3f), indicating the amplitudes of temperature and amount effects on millennial timescale. The millennial

climatology is derived from all the 23 time slices (using the last 40 years of model output). We can see the correlations on this timescale are, overall, more significant than on interannual timescale, but much less than on seasonal timescale. It is also shown that neither effect is significant over the eastern China (105E-120E, 22N-42N) including North China, central China, and South China. Hence, the model do not support strong link of stalagmite $\delta^{18}$O records found surrounding this area with local either temperature or precipitation. Some exception areas are in SCS, showing negative correlations with temperature and

precipitation, as well as northeast China, showing positive correlation with precipitation.

The millennial variability study here is consistent with previous modeling studies. In a serious of snapshot simulations for the Holocene (LeGrande and Schmidt, 2009) and in an idealized millennial variability (hosing) experiment (Pausata et al., 2011), it is suggested that the millennial variability of cave $\delta^{18}$O in southeastern China do not represent that of local monsoon rainfall; rather, it reflects the change of the upstream moisture transport and the variability in the upstream Indian Ocean and South

Asian monsoon region. However, because of the coherent continental scale monsoon response between the upstream region and rainfall in northern China, Liu et al. (2014b) suggest that the Chinese cave records is still a good indicator of the millennial variability of the southerly monsoon wind, and the monsoon rainfall in northern China, although not of local monsoon rainfall over the cave sites in southeastern China. This suggests that one needs to cautious in interpreting the water isotope records in





the proxy records. Instead of using the modern day seasonal cycle as a simple analogy, the interpretation of the $\delta^{18}$O variability of millennial time scales differs in different regions. A credible interpretation requires a better understanding of the dynamics of the monsoon rainfall as well as the mechanism responsible for the water isotope variability.

## 4    Conclusions

We have examined the temperature and amount effects on seasonal, interannual, and millennial timescales in the East Asia in an isotope-enable atmospheric GCM. It is found that, the two effects hold well on the seasonal timescale, with the temperature effect dominating in northern China and amount effect dominating in very southern China. However, neither temperature effect nor amount effect is robust for interannual variability. Neither effect is strong on millennial timescale either, albeit somewhat stronger than that at interannual timescale.

The cause for the different isotope-climate relations at different time scales remain to be studied in the future. Tentatively, we speculate, part of the reason that temperature effects and amount effect are robust for seasonal cycle is because the seasonal cycle has a large variance and therefore perhaps large signals. Fig 4 shows the correlations of $\delta^{18}$O-temperature and $\delta^{18}$O-precipitation (in vectors) in various regions on three distinct timescales in China, with respect to the local variances of temperature and precipitation. It is shown that the temperature and amount effects are robust on seasonal timescales (with cor-

relation coefficients mostly over 0.6). Furthermore, the temperature effect tend to be dominant for regions of large temperature variability in seasonal cycle (northern China), while the amount effects (negative correlation) tends to be robust in regions of large rainfall seasonal cycle (southern China). This is consistent with the observations from GNIP networks in China (Fig 4a). In comparison, the temperature and amount effects are weak on interannual and millennial timescales, partly because of their small variances of temperature and precipitation and induced small signal-to-noise ratio.

Our model shows that the classic empirical relations between oxygen isotopes and local temperature and precipitations as derived from seasonal cycle do not apply to other time scales in general. Hence, one should be cautious in using present isotope-climate relation for long-term paleoclimate reconstructions. For example, although the oxygen isotope variability in southeastern China is not caused directly by local precipitation variability, it could still be correlated with the intensity of the East Asian summer monsoon, which is correlated with the intensity of southerly monsoon wind, the moisture transport,

and the associated large-scale rainfall response (Liu et al., 2014b). Therefore, we suggest that for paleoclimate studies one should focus more on the large-scale circulations associated with the stable water isotopes rather than relying heavily on local isotope-climate relations.



*Author contributions.* X. Wen and Z. Liu conceived the study and wrote the paper, D. Noone developed the module of water stable isotope in CAM3, X. Wen and E. Brady performed the slice simulations, Z. Chen contributed to interpret GNIP observations, Q. Zhu and J. Guan performed the analysis. All authors discussed the results and provided inputs to the paper.

*Acknowledgements.* This work is supported by National Science Foundation of China (Grand No. 41130105, 41130962, and 41005035), Beijing Young Elite Foundation (YETP0005), and by NSF C2P2 and DOE SciDac.





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



**Figure captions**

**Figure 1**

Map of the stations in GNIP network over China (a) and the availability of $\delta^{18}$O data during the period 1961-2005 (b). The valid monthly mean $\delta^{18}$O records are marked in gray. In particular, a complete one-year data is highlighted in black. The eight regions defined in model

outputs are demonstrated in (c).

**Figure 2**

Seasonal cycles of $\delta^{18}$O in precipitation (color lines), surface temperature (solid lines), and precipitation (dashed lines) over eight regions in China. The left column shows typical profiles from eight stations within GNIP network, whereas the right column shows model results averaged over the corresponding region defined in Fig 1c.

10 **Figure 3**

The point-to-point correlation coefficients between $\delta^{18}$O and temperature (left column) or precipitation (right column) on three timescales. The seasonal timescale (a and b) uses the last 40-year monthly data from the 00ka snapshot, excluding MAM (spring) and SON (fall) months to avoid the noise. The interannual timescale (c and d) uses 40-year annual mean $\delta^{18}$O weighted with precipitation and DJF temperature and JJA precipitation from the 00ka slice. The millennial timescale (e and f) uses the climatology data from 23 snapshots (22ka, 21ka. . . 01ka,

00ka). All the statistically insignificant areas under 95% confidence level are left as blank.

**Figure 4**

Temperature effect and amount effect (vectors) of $\delta^{18}$O records in model outputs (open markers) and GNIP observations (solid circle) over eight regions (colors) in China with respect to the variances of temperature and precipitation on three timescales. The cyan shaded part in (a) is enlarged as (b). Each of the timescales shows distinct regime in this diagram.



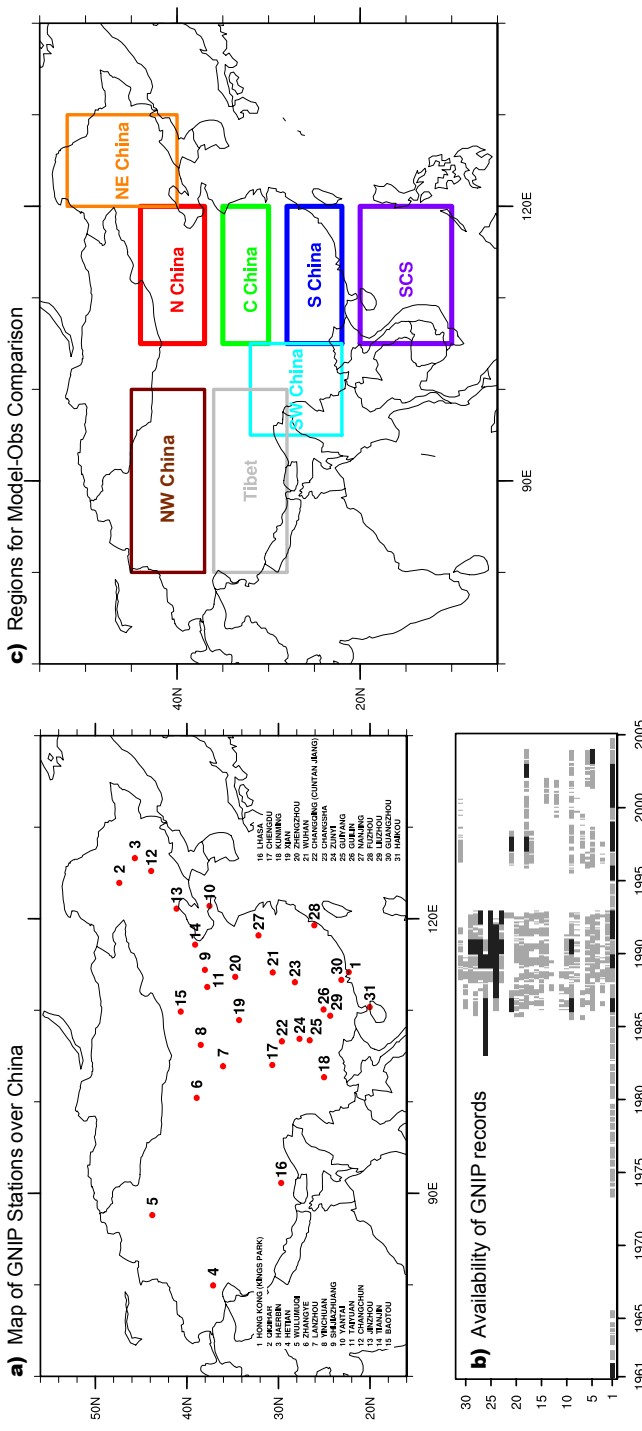

**Figure 1.** Map of the stations in GNIP network over China (a) and the availability of $\delta^{18}$O data during the period 1961-2005 (b). The valid monthly mean $\delta^{18}$O records are marked in gray. In particular, a complete one-year data is highlighted in black. The eight regions defined in model outputs are demonstrated in (c).





**Figure 2.** Seasonal cycles of $\delta^{18}$O in precipitation (color lines), surface temperature (solid lines), and precipitation (dashed lines) over eight regions in China. The left column shows typical profiles from eight stations within GNIP network, whereas the right column shows model results averaged over the corresponding region defined in Fig 1c.



**Figure 3.** The point-to-point correlation coefficients between $\delta^{18}O$ and temperature (left column) or precipitation (right column) on three timescales. The seasonal timescale (a and b) uses the last 40-year monthly data from the 00ka snapshot, excluding MAM (spring) and SON (fall) months to avoid the noise. The interannual timescale (c and d) uses 40-year annual mean $\delta^{18}O$ weighted with precipitation and DJF temperature and JJA precipitation from the 00ka slice. The millennial timescale (e and f) uses the climatology data from 23 snapshots (22ka, 21ka... 01ka, 00ka). All the statistically insignificant areas under 95% confidence level are left as blank.



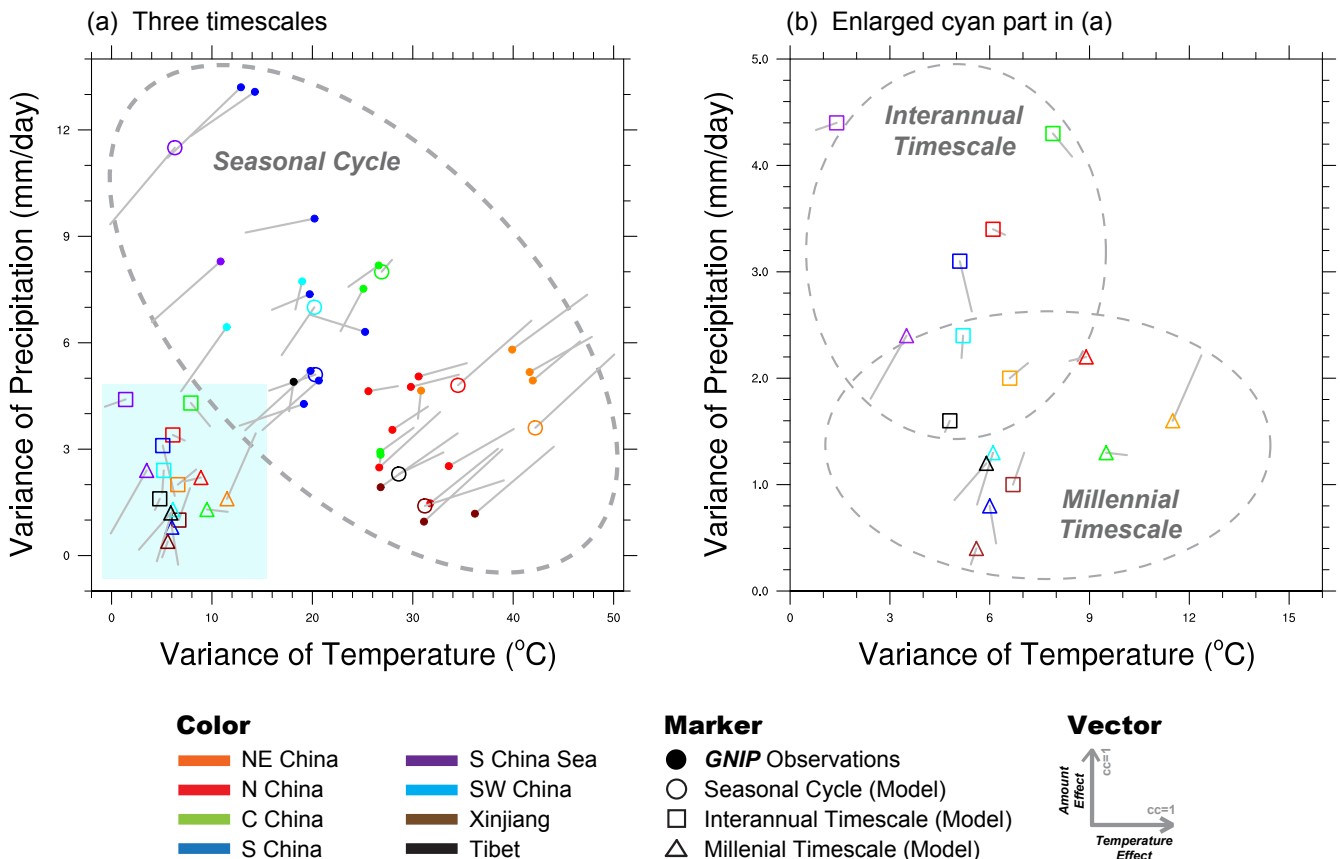

**Figure 4.** Temperature effect and amount effect (vectors) of $\delta^{18}O$ records in model outputs (open markers) and GNIP observations (solid circle) over eight regions (colors) in China with respect to the variances of temperature and precipitation on three timescales. The cyan shaded part in (a) is enlarged as (b). Each of the timescales shows distinct regime in this diagram.