# Peer review of "Modeling precipitation $\delta^{18}$ O variability in East Asia since the Last Glacial Maximum: temperature and amount effects across different time scales"

_Climate of the Past, 2016_

## Referee Comment (RC1) · Anonymous Referee #1 · 16 May 2016

General comments: This study looks at how d18Oprecip relates to temperature and precipitation over the East Asian monsoon regions on time scales of variability from seasonal to millennial using an isotope-enabled atmosphere-only climate model. The authors find interesting results that they say mean that d18O speleothem records should be interpreted with caution. I am very supportive of such modelling work to help the palaeoclimate community to better understand how isotopic data might be interpreted. I did though find the paper was quite descriptive and short in its explanations, and would benefit from the addition of further investigation into all the main points they made in order to improve the mechanistic understanding (see comments below). I recommend this be done before the manuscript can be published.

Specific corrections: Abstract Line 1: change "Water isotope in precipitation has played a key role" to "Water isotopes in precipitation have played a key role"

Abstract line 1: add references to support the first statement

Abstract line 5: Although I realise 'thru' is sometimes used for ranges esp. in American English, I would recommend changing "22ka thru 00ka using an isotope-enable AGCM" to "22 ka to 0 ka using an isotope-enabled atmospheric global circulation model (AGCM)".

Abstract line 7: "Our study confirms the robustness of the temperature and amount effects on the seasonal cycle over China" – does this statement refer just to the present day? Please add to the text.

Abstract line 8: "our analysis does not show significant temperature and amount effects over China on millennial and interannual timescales" – do you mean no significant change, or neither is significantly dominant?

Introduction page 1 line 13: "Sturm et al., 2010; Noone, 2008" – add an 'e.g.' and perhaps reference to some of the older earlier pioneering papers on this

Introduction page 1 line 15: ""local temperature effect", whereas the $\delta 18O$-precipitation relationship in the tropics and low latitudes tends to be associated with the "amount" effect" – I would be keen to see a small amount of explanation of these terms for any readers who might be relatively new to the subject.

Page 2 line 9 Change "East Asia locates at the transition zone" to " East Asia is located at. . ."

Page 2 line 10 "still remains as a great controversy" – delete "as"

Page 2 line 11 and all other instances of "isotope-enable GCM" change to "isotope-enabled GCM"

Page 2 line 20: delete "proxies".

Page 2 line 26: "These experiments are forced by the realistic green house gases (GHGs) concentrations, orbital parameters, land ice sheet and land-ocean mask" – are these all the same boundary conditions as used in the Liu et al. papers, as well as the SSTs/sea-ice?

Page 2 line 30: "1.6‰ (22ka) to 0.5‰ (0ka)" - please say/reference where you have derived the values from, which have then been linearly interpolated, if I understand right.

Page 3 line 1: add reference for the GNIP data

Page 3 line 3: change "This dataset has sufficient spatial coverage. But majority of. . ." to "This dataset has sufficient spatial coverage but the majority of. . ."

Page 3 line 4: change "there is only12 stations" to "there are only12 stations"

Page 3 line 4: change 'showing' to 'shown'

Page 3 line 18: "For each region, the modeled seasonal cycle are derived from" change to "For each region, the modeled seasonal cycle is derived from"

Figure 2: I know the names of the GNIP stations used are included in the plot, but is it also possible to add in the number that corresponds to the number of the site in figure 1.

Figure 2: The comparison of the left and right hand graphs is slightly improved as the y-axes have different limits. While I see that this is to maximize the details, it would be easier to make comparisons if the scales were the same. – Actually I realize this is mentioned on page 4 in the penultimate paragraph.

Page 3 line 20: why only use the 'GNIP station that has the longest records in that region' in the comparison in figure 2. Have you checked whether there is good correspondence between the one record chosen for each region and the other shorter records in the region? I.e. is each particular record indicative of the overall pattern in

the region? Otherwise it seems insufficient reason to choose a particular record based on its length, or say why a longer record is better – e.g. to reduce the impact of interannual variability? Related to this – page 4 paragraph around line 25 – states that the d18O values have somewhat different magnitudes although the phase is a good match with the data, however, there are similar differences in precip and temperature between model and data (as I'd imagine with most models), which might be worth also pointing out in this paragraph.

Page 4 line 8: discusses that the d18O signal from the model in southern China doesn't replicate the seasonal pattern in the data and suggests a resemblance to the 'third mode' as discussed in the following paragraph. However, no mention is made of the fact that the seasonality of precipitation isn't quite right over S China either and how this could influence the mismatch between the model and data d18O.

Page 4 line 18: change 'implications to the interpretation' to 'implications for the interpretation'.

Page 4 line 20: 'Thus, we would suggest that one should NOT interpret the $\delta$18O records around this region simply as the monsoon rainfall amount.' One could also suggest that the boundaries between these different regions could change significantly over time (through glacial-interglacial cycles fro example). It would be useful if the authors could say something regarding this uncertainty and the implications for interpretation of palaeo-isotopic records.

Page 4 line 31: 'This distinctively different three regions' change to 'These three distinctively different regions'

Page 5, line 10-15 These lines contain a suggestion of why south Asia and East Asia show different correlations between d18O and temp/precip on interannual timescales, but not enough detail to understand the mechanisms for this beyond them having different moisture sources. I suggest a clearer and more detailed explanation is necessary here.

Page 5 line 20: 'using the last 40 years of model output' do you mean where each of the 23 year time slices provides one time point that is the average of the last 40 model years of that simulation. Text could be a bit clearer.

Page 5 line 25 onwards: it is an interesting result that millennial-scale variability in d18O doesn't reflect high significance in correlation with local temperature or precip. In line with other studies, the authors suggest that d18O over East Asia could be influenced by upstream moisture transport from the Indian monsoon region (similar to Pausata et al). However, they do not investigate this any further in their model so we do not learn as much as we could about what mechanisms are important factors here. The authors have all the data at their disposal and so could look at e.g. correlation at the millennial time-scale of Indian monsoon temp/precip/d18O with d18O over China, and variability in the southerly monsoon winds etc. I would like to see the authors examine what is driving their millennial scale variation further.

Page 5: line 25: Related to the above point, does the seasonality of precipitation/temperature/d18O change much in the different locations in these 23 time slices? Do the seasonal correlations, interpreted as d18O being affected by the temperature effect in the north and the precipitation effect in the south still hold for the same locations or do the boundaries change from glacial to interglacial time slices?

––––––––––––––––––––––––––

---

## Referee Comment (RC2) · Anonymous Referee #2 · 17 May 2016

The authors used a list of time slice experiments by an isotope-enabled GCM to evaluate the changes in precipitation ïĄď18O on various timescales. It is an interesting work and might give some insights for the interpretation of stalagmite ïĄď18O, especially for the paleoclimate reconstructions in Asia. I do not know whether these experiments are the same as those in Liu et al. 2014 QSR or not. The authors should clarify this in the section of model description. These experiments are no doubt useful for exploring the interpretation of the precipitation ïĄď18O over the East Asian on different time scale. However, I am afraid that the present experiment design is not reasonable enough for examining the changes in ïĄď18O, especially on the seasonal and inter-annual timescales. The present 0Ka experiment may neglect some major changes in

boundary conditions and can not directly compare to the modern GNIP observations. Are the greenhouse gases and sea surface temperature kept constant? Why do the authors not employ the observed GHG and SST to force the atmosphere model? This experiment is necessary and do not need much time. I strongly recommend to add this experiment and to reanalyze the results.

The authors use a series of time-slice experiments for the last 22 ka to evaluate the "temperature effect" and "amount effect" on millennial time scale in different regions in East Asia. I think that the author should present the long-term changes of precipitation ïĄď18O in these model simulations and compare them with the proxy records. If the outputs of these experiments capture the variations in the proxy time series, then it's robust to test the interpretation of the precipitation ïĄď18O on millennial time scale by using the model simulation. Otherwise, the bias in the model itself will mask the real processes which affect the precipitation ïĄď18O changes. This is fundamental to the model simulation. The authors must cross check the model outputs with the real observations and then come to the conclusion.

As shown in figure 3, the authors correlate the annual mean ïĄď18O weighted with precipitation to the DJF temperature and JJA precipitation on the interannual and millennial time scales (panel c-f) and then use this statistic result to argue the "amount effect" and "temperature effect". This is totally wrong! Because the annual mean temperature may not change the same way as the DJF temperature, and also the varied precipitation seasonality (as shown in figure 2) in different regions may deny the dominant contribution of summer precipitation to the annual precipitation.

Page 1 line 19, the citation of Yuan et al., 2004 is wrong. It presents the speleothem ïĄď18O record from southern China.

———————————————

---

## Author Comment (AC1) · 21 Jun 2016

We would like to appreciate the anonymous reviewer for his/her helpful suggestions. The original comment (Q) and our response (A) are as follows:

Q: Specific corrections: Abstract Line 1: change "Water isotope in precipitation has played a key role" to "Water isotopes in precipitation have played a key role"

A: Done.

Q: Abstract line 1: add references to support the first statement

A: Since it is not recommended to add reference in the abstract part for format reason, the references are included in the first statement of the main text.

Q: Abstract line 5: Although I realise 'thru' is sometimes used for ranges esp. in American English, I would recommend changing "22ka thru 00ka using an isotope-enable AGCM" to "22 ka to 0 ka using an isotope-enabled atmospheric global circulation model (AGCM)".

A: Thanks. It is changed.

Q: Abstract line 7: "Our study confirms the robustness of the temperature and amount effects on the seasonal cycle over China" – does this statement refer just to the present day? Please add to the text.

A: We have checked the two effects (temperature effect and amount effect) on seasonal cycle timescale in present day and the past slices in the model outputs. The supplementary Fig 1 shows an example for North China. We can see the conclusion based on the present day conditions does not change much in the past 22,000 years. But given lack of the observations at seasonal timescale for the past, we would prefer to make the conclusion, in this paper, just for the present day, as compared to the observed monthly data from GNIP. Thanks, we added "in the present climatic conditions" in the text to explicitly clarify this point.

Q: Abstract line 8: "our analysis does not show significant temperature and amount effects over China on millennial and interannual timescales" – do you mean no significant change, or neither is significantly dominant?
A: The latter one. This sentence was changed to "our analysis shows that neither temperature nor amount effects is significantly dominant over China on millennial and interannual timescales".

Q: Introduction page 1 line 13: "Sturm et al., 2010; Noone, 2008" – add an 'e.g.' and perhaps reference to some of the older earlier pioneering papers on this

A: Thanks, it is changed. Also, we added some pioneering papers, such as Dans-gaard1964, Grootes1993, Cuffey1995, and Salamatin1998.

Q: Introduction page 1 line 15: ""local temperature effect", whereas the d18Oprecipitation relationship in the tropics and low latitudes tends to be associated with the "amount effect" – I would be keen to see a small amount of explanation of these terms for any readers who might be relatively new to the subject.

A: Thanks! Good point! We added "a positive correlation between d180 in precipitation and the temperature of ambient air (warmer air provide more energy to rain out 180rich water vapor)" to briefly describe the temperature effect, and added "a negative correlation between d180 in precipitation and the accumulated total rainfall at local and upstream regions (stronger tropical precipitation leave less d180 in vapors transported to the subtropics and low latitudes)" to briefly introduce the amount effect in the text.

Q: Page 2 line 9 Change "East Asia locates at the transition zone" to "East Asia is located at"

A: Done.
Q: Page 2 line 10 "still remains as a great controversy" - delete "as"

A: Deleted.

Q: Page 2 line 11 and all other instances of "isotope-enable GCM" change to "isotope enabled GCM"

A: Many thanks! We changed all the instances of "isotope-enable" in the text to "isotope-enabled".

Q: Page 2 line 20: delete "proxies".

A: Deleted.

Q: Page 2 line 26: "These experiments are forced by the realistic green house gases (GHGs) concentrations, orbital parameters, land ice sheet and land-ocean mask" – are these all the same boundary conditions as used in the Liu et al. papers, as well as the SSTs/sea-ice?

A: Yes, all the boundary conditions are the same as used in Liu et al. (2009).

Q: Page 2 line 30: "1.6 permil (22ka) to 0.5 permil (0ka)" - please say/reference where you have derived the values from, which have then been linearly interpolated, if I understand right.

A: They came from other's work. References added: Schrag et al. (1996) for 22ka and Hoffmann et al. (1998) for 0ka.
Q: Page 3 line 1: add reference for the GNIP data

A: Reference added: Schotterer and Oldfield (1996).

Q: Page 3 line 3: change "This dataset has sufficient spatial coverage. But majority of" to "This dataset has sufficient spatial coverage but the majority of"

A: Done.

Q: Page 3 line 4: change "there is only12 stations" to "there are only12 stations" A: Done.

| Q: Page 3 | line 4: | change | 'showing' | to | 'shown' |
|-----------|---------|--------|-----------|----|---------|
|           |         |        |           |    |         |

A: Done.

Q: Page 3 line 18: "For each region, the modeled seasonal cycle are derived from" change to "For each region, the modeled seasonal cycle is derived from"

A: Done.

Q: Figure 2: I know the names of the GNIP stations used are included in the plot, but is it also possible to add in the number that corresponds to the number of the site in figure 1.

A: Do you suggest us to add a table for 31 GNIP stations? Great idea! We collected their basic information and would like to add a table in the revised manuscript, as shown
in the attached file.

Q: Figure 2: The comparison of the left and right hand graphs is slightly improved as the y-axes have different limits. While I see that this is to maximize the details, it would be easier to make comparisons if the scales were the same. – Actually I realize this is mentioned on page 4 in the penultimate paragraph.

A: Many many thanks! Good point! The left and right columns share the same y-scale in the revised version. You may check this in supplementary Figure 2 in advance.

Q: Page 3 line 20: why only use the 'GNIP station that has the longest records in that region' in the comparison in figure 2. Have you checked whether there is good correspondence between the one record chosen for each region and the other shorter records in the region? I.e. is each particular record indicative of the overall pattern in the region? Otherwise it seems insufficient reason to choose a particular record based on its length, or say why a longer record is better – e.g. to reduce the impact of interannual variability? Related to this – page 4 paragraph around line 25 – states that the d18O values have somewhat different magnitudes although the phase is a good match with the data, however, there are similar differences in precip and temperature between model and data (as I'd imagine with most models), which might be worth also pointing out in this paragraph.

A: Yes, we did so to reduce the uncertainty of interannual variability. Given the feature of GNIP data's discontinuity (A. many missing within a year; B. observed years is so short, commonly no more than 10 years from 1985 to 1993), as shown in Fig 1(b), the longer records one station has, the better quality in representing reliable seasonal cycle of d18O/temperature/precipitation. The supplementary Figure 3 shows an example for NE China: QIQIHAR station, the one used in our study that has longest records among
the 4 stations in this region, gives a highly consistent seasonal cycle as the mean. We checked this point for other regions and the same thing happened.

Q: Page 4 line 8: discusses that the d18O signal from the model in southern China doesn't replicate the seasonal pattern in the data and suggests a resemblance to the 'third mode' as discussed in the following paragraph. However, no mention is made of the fact that the seasonality of precipitation isn't quite right over S China either and how this could influence the mismatch between the model and data d18O.

A: Thanks for this helpful comment. We rewrite the sentence like this: "Instead, the modeled d18O in southern China exhibits a double maximum in spring and fall partly due to the incorrect seasonality of precipitation with its maximum occurred near May-June. The model cannot well reproduce the climatology in this region as it slightly resembles the third mode to be discussed next."

Q: Page 4 line 18: change 'implications to the interpretation' to 'implications for the interpretation'.

A: Done.

Q: Page 4 line 20: 'Thus, we would suggest that one should NOT interpret the d180 records around this region simply as the monsoon rainfall amount.' One could also suggest that the boundaries between these different regions could change significantly over time (through glacial-interglacial cycles fro example). It would be useful if the authors could say something regarding this uncertainty and the implications for interpretation of palaeo-isotopic records.

A: Thanks! The changes of Asian monsoon advancing/retreating during glacial-
interglacial cycle are much smaller than the amplitude of seasonal cycle of d18O/T/P. Thus, the position of this transition region is robust across glacial-interglacial cycle and change little. We made the supplementary Figure 4, which shows the model results from other time slices than 00ka (Fig 3a and b). We can see the model suggests a robust and almost stationary "blank region" over the central China.

Q: Page 4 line 31: 'This distinctively different three regions' change to 'These three distinctively different regions'

A: Done.

\_\_\_\_

Q: Page 5, line 10-15 These lines contain a suggestion of why south Asia and East Asia show different correlations between d18O and temp/precip on interannual timescales, but not enough detail to understand the mechanisms for this beyond them having different moisture sources. I suggest a clearer and more detailed explanation is necessary here.

A: Thanks. This point is also closely related to the penultimate question about the relationship between Chinese d18O and Indian monsoon. Please find our response and associated reference in that part together. We will add more words to discuss their relationship in the revised text.

Q: Page 5 line 20: 'using the last 40 years of model output' do you mean where each of the 23 year time slices provides one time point that is the average of the last 40 model years of that simulation. Text could be a bit clearer.

A: Thanks. This sentence is changed to "The millennial climatology is derived from each time slice by averaging the last 40 years out of 50-year raw results."
Q: Page 5 line 25 onwards: it is an interesting result that millennial-scale variability in d18O doesn't reflect high significance in correlation with local temperature or precip. In line with other studies, the authors suggest that d18O over East Asia could be influenced by upstream moisture transport from the Indian monsoon region (similar to Pausata et al). However, they do not investigate this any further in their model so we do not learn as much as we could about what mechanisms are important factors here. The authors have all the data at their disposal and so could look at e.g. correlation at the millennial time-scale of Indian monsoon temp/precip/d18O with d18O over China, and variability in the southerly monsoon winds etc. I would like to see the authors examine what is driving their millennial scale variation further.

A: Thanks for great suggestion! In another paper (Liu and Wen, 2014) with focus on the summer monsoon dynamics, we investigated the variability of d18O on orbital timescale over China, and its relations to Indian precipitation as well as the southerly monsoon winds. Basically, Chinese d18O highly correlated with Indian d18O and precipitation, suggesting a reliable dynamic link between Indian precipitation and Chinese isotope records through amount effect. On the other hand, East Asian monsoon is also influenced by the western North Pacific and South China Sea rather than just the Indian Ocean. The complex circulation determines China's nature having multiple modes of precipitation and d18O. We will add more discussion for this part.

Reference: Liu, Z and X Wen et al., 2014: Chinese cave d18O records representing East Asia summer monsoon, Quan. Sci. Rev., 83, 115-128.

Q: Page 5: line 25: Related to the above point, does the seasonality of precipitation/ temperature/d18O change much in the different locations in these 23 time slices? Do the seasonal correlations, interpreted as d18O being affected by the temperature effect in the north and the precipitation effect in the south still hold for the same locations or
do the boundaries change from glacial to interglacial time slices?

A: Similarly, please take a look at the supplementary Figure 4 for the spatial distribution of temperature/amount effects on seasonal timescale across 20ka, 15ka, 10ka, 5ka, and 0ka. It is shown that the pattern, temperature dominating north and precipitation dominating south, does not change much in the last 20,000 years. You may also check the supplementary Figure 1 for the details of d18O/T/P seasonal cycle over North China, as an example, across the last 22,000 years.

---

## Author Comment (AC2) · 21 Jun 2016

We would like to appreciate the anonymous reviewer for his/her helpful comments. The original comment (Q) and our response (A) are as follows:

—

Q: The authors used a list of time slice experiments by an isotope-enabled GCM to evaluate the changes in precipitation d18O on various timescales. It is an interesting work and might give some insights for the interpretation of stalagmite d18O, especially for the paleoclimate reconstructions in Asia. I do not know whether these experiments are the same as those in Liu et al. 2014 QSR or not. The authors should clarify

this in the section of model description. These experiments are no doubt useful for exploring the interpretation of the precipitation d18O over the East Asian on different time scale. However, I am afraid that the present experiment design is not reasonable enough for examining the changes in d18O, especially on the seasonal and interannual timescales. The present 0Ka experiment may neglect some major changes in boundary conditions and can not directly compare to the modern GNIP observations. Are the greenhouse gases and sea surface temperature kept constant? Why do the authors not employ the observed GHG and SST to force the atmosphere model? This experiment is necessary and do not need much time. I strongly recommend to add this experiment and to reanalyze the results.

A: Yes, the numerical experiments are the same as those in Liu et al. (QSR, 2014). We added a sentence in section 2 to clarify this point. In Liu's paper (2014), we discussed the dynamic linkage between Chinese d18O and East Asian summer monsoon, whereas in this paper we would like to discuss the robustness of interpreting d18O records in terms of two effects on three different timescales: seasonal, interannual, and millennial. At the early stage of this work, we planned to focus on 4 timescales: millennial, interdecadal, interannual, and seasonal. But the big problem is lack of observed d18O record on interannual-to-interdecadal timescales. Most of the records from GNIP network have no more than 8-year (1985-1993) consecutive history (See Fig 1b). Thus, we removed the "interdecadal" and focus on the remaining three, among them the "interannual timescale", relatively, could be lacking data most. In general, the interannual variability of d18O or other variables include two sources: climate system internal variance and responses to external forcing. The observed d18O records are too short to reliably account for both. Our 00ka slice was driven by 1950 boundary conditions and was integrated for 50 years, which is able to provide more samples on interannual timescale than GNIP for internal variability problem, but for forcing-response problem. This is the shortage of current experiments. Thank the reviewer's kind suggestion, we would like to develop a number of AMIP-type ensemble experiments in the next phase to investigate the response and sensitivity of water isotopes to external

forcing, like ENSO or global warming.

—

Q: The authors use a series of time-slice experiments for the last 22 ka to evaluate the "temperature effect" and "amount effect" on millennial time scale in different regions in East Asia. I think that the author should present the long-term changes of precipitation d18O in these model simulations and compare them with the proxy records. If the outputs of these experiments capture the variations in the proxy time series, then it's robust to test the interpretation of the precipitation d18O on millennial time scale by using the model simulation. Otherwise, the bias in the model itself will mask the real processes which affect the precipitation d18O changes. This is fundamental to the model simulation. The authors must cross check the model outputs with the real observations and then come to the conclusion.

A: Thanks for this great comment! We compared the model results (d18O, precipitation, and meridional winds) with proxy data in Liu et al. (QSR, 2014). It is shown that the model successfully reproduces the observed orbital and millennial variability as compared to multiply proxies and generates reliable monsoon-associated anomalous circulation (in Liu et al., QSR, 2014, Fig 2e shows model results by comparing with d18O proxy and V winds; Fig 2f and 2g shows the comparison of modeled precipitation and other lake sediment proxies). This forms the solid base for present investigation. We attached this paper as a supplementary file.

Reference: Liu, Z and X Wen et al., 2014: Chinese cave d18O records representing East Asia summer monsoon, Quan. Sci. Rev., 83, 115-128.

—

Q: As shown in figure 3, the authors correlate the annual mean d18O weighted with precipitation to the DJF temperature and JJA precipitation on the interannual and millennial time scales (panel c-f) and then use this statistic result to argue the "amount

effect" and "temperature effect". This is totally wrong! Because the annual mean temperature may not change the same way as the DJF temperature, and also the varied precipitation seasonality (as shown in figure 2) in different regions may deny the dominant contribution of summer precipitation to the annual precipitation.

A: Many thanks! The observed d18O records in speleothem, fundamentally, reflects precipitation-weighted annual mean value. For Asian monsoon region, it could be considered that these cave d18O records mostly take summertime rather than wintertime information. However, the water stable isotope's temperature effect mainly occur over high latitudes in winter, whereas the amount effect mainly occur over tropics in summer. Thus, in the discussion version of the manuscript, we compared prec.-weighted d18O with DJF temperature and JJA precipitation. Here, we re-examine this problem for millennial (supplementary Figure 1) and interannual (supplementary Figure 2) timescales. It is shown that the prec.-weighted annual mean of temperature and precipitation could be the appropriate variables accounting for temperature effect and amount effect. They are even more reasonable than equal-weighted annual mean by emphasizing rain-season footprint. Also, the varied seasonality of precipitation is implicitly considered. We will replace the corresponding plots in Figure 3 and modify the text in the revised manuscript.

We further investigate the robustness of interannual patter (as above) of temperature/amount effects across the past 22,000 years (say, 20ka, 15ka, 10ka, 5ka, and 0ka), as shown in the supplementary Figure 3. It is shown that the weak correlation (the blank) region does not change much, suggesting the conclusion that one should be very cautious in interpreting d18O records for this area on interannual timescale still remain.

—

Q: Page 1 line 19, the citation of Yuan et al., 2004 is wrong. It presents the speleothem d18O record from southern China.

A: Thanks, we moved this item to speleothem part.

Please also note the supplement to this comment:
http://www.clim-past-discuss.net/cp-2016-2/cp-2016-2-AC2-supplement.pdf
———————————————

**Millennial Timescale**

[Figure]

**Fig. 1.** Selection of temperature/precipitation associated with precipitation-weighted annual mean d18O through temperature/amount effect on millennial timescale.

[Figure]

**Fig. 2.** Selection of temperature/precipitation associated with precipitation-weighted annual mean d18O through temperature/amount effect on interannual timescale.

[Figure]

[Figure]

**Fig. 3.** The variation of spatial pattern of temperature/amount effect on interannual timescale across the past 20,000 years.

**Supplement:**

Quaternary Science Reviews 83 (2014) 115–128

Contents lists available at ScienceDirect

**Quaternary Science Reviews**

journal homepage: www.elsevier.com/locate/quascirev

[Figure]

**Chinese cave records and the East Asia Summer Monsoon**

[Figure]

Zhengyu Liu [a,b,*], Xinyu Wen [a], E.C. Brady [c], B. Otto-Bliesner [c], Ge Yu [d], Huayu Lu [e],
Hai Cheng [f,g], Yongjin Wang [h], Weipeng Zheng [i], Yihui Ding [j], R.L. Edwards [g], Jun Cheng [k],
Wei Liu [b], Hao Yang [l]

[a] Laboratory Climate, Ocean and Atmospheric Studies, School of Physics, Peking University, Beijing 100871, China
[b] Center for Climatic Research, University of Wisconsin–Madison, Madison, WI 53706, USA
[c] Climate and Global Dynamics Division, National Center for Atmospheric Research, Boulder, CO 80307-3000, USA
[d] State Key Laboratory of Lake Science and Environment, Nanjing Institute of Geography and Limnology, Chinese Academy of Sciences, Nanjing 210008, China
[e] School of Geographic and Oceanographic Sciences, Institute for Climate and Global Change Research, Nanjing University, Nanjing 210093, China
[f] Institute of Global Environmental Change, Xi'an Jiaotong University, Xi'an 710049, China
[g] Department of Earth Sciences, University of Minnesota, Minneapolis, MN 55455, USA
[h] College of Geography Science, Nanjing Normal University, Nanjing 210097, China
[i] State Key Laboratory of Numerical Modeling for Atmospheric Sciences and Geophysical Fluid Dynamics, Institute of Atmospheric Physics, Chinese Academy of Sciences, Beijing 100029, China
[j] Laboratory of Climate Studies, China Meteorological Administration, Beijing 100081, China
[k] College of Oceanography, Nanjing University of Information Science and Technology, Nanjing 210044, China
[l] Key Laboratory of Meteorological Disaster, Nanjing University of Information Science and Technology, Nanjing 210044, China

**ARTICLE INFO**

Article history:
Received 28 February 2013
Received in revised form
16 October 2013
Accepted 21 October 2013
Available online 27 November 2013

Keywords:
East Asia summer monsoon
Oxygen isotope
Climate modeling
Isotope modeling

**ABSTRACT**

Speleothem records in southeastern China provide key evidence for past environmental changes. However, the climatic interpretation of these proxies has remained a great controversy. Earlier work interprets the cave $\delta^{18}O$ signal associated with regional rainfall of the East Asia Summer Monsoon (EASM) or monsoon rainfall upstream of China. Recent isotope modeling supports the latter but show little correspondence between the precipitation $\delta^{18}O$ and rainfall in China. Here, we examine the evolution of the climate and precipitation $\delta^{18}O$ for the last 21,000 years in models and observations. Recognizing the regional difference of the EASM rainfall, we propose an interpretation of the Chinese $\delta^{18}O$ record that reconciles its representativeness of EASM and its driving mechanism of upstream depletion. The $\delta^{18}O$ records do represent the intensity of the EASM system. The monsoon intensity is best characterized by enhanced southerly monsoon winds, which correlate strongly with negative $\delta^{18}O$ over China and enhanced monsoon rainfall in northern China, as well as the continental scale Asian monsoon rainfall response in the upstream regions.

© 2013 Elsevier Ltd. All rights reserved.

**1. Introduction**

Speleothem oxygen isotope records in southeastern China caves ($\delta^{18}O_c$, for $\delta^{18}O$ in the cave) (Fig. 1a) show large variability coherent with climate records in the Arctic (Fig. 2d and e) (Wang et al., 2001; Yuan et al., 2004; Hu et al., 2008; Cheng et al., 2009, 2012), providing key evidence for past environmental changes. However, the climatic interpretation of these proxies has remained a great controversy. Early work interprets these cave $\delta^{18}O_c$ signals as a proxy for regional rainfall of the East Asia Summer Monsoon (EASM), with higher EASM rainfall corresponding to depleted (more negative) precipitation oxygen isotope $\delta^{18}O_p$ (for $\delta^{18}O$ in precipitation). This relationship is based on two possible mechanisms. Simple Rayleigh fractionation calculations (e.g. Gat, 1996) demonstrate that the cave observations are consistent with changes in rainfall integrated between tropical ocean sources and Chinese cave sites (the upstream depletion mechanism) (Yuan et al., 2004) and rainfall integrated between two Chinese cave sites (Hu et al., 2008). Alternately, as today's EASM rainfall is anomalously low in $\delta^{18}O$, a higher proportion of regional EASM rainfall in annual totals would result in more negative $\delta^{18}O_c$ (Cheng et al., 2009). These proposed mechanisms have been discussed with

* Corresponding author. Center for Climatic Research, University of Wisconsin–Madison, 1225 W. Dayton Street, Madison, WI 53706, USA. Tel.: +1 608 262 0777; fax: +1 608 263 4190.

E-mail address: zliu3@wisc.edu (Z. Liu).

0277-3791/$ – see front matter © 2013 Elsevier Ltd. All rights reserved.
http://dx.doi.org/10.1016/j.quascirev.2013.10.021

[Figure]

**Fig. 1.** Speleothem δ$^{18}$O records from (a) the East Asia monsoon region and (b) the Indian monsoon region. These records show a coherent variability within China and, moreover, with those in India. In panel (a), (A) Summer (JJA) mean Insolation at 65°N (Laskar et al., 2004). (B) The Jiuxian record (33°34′ N, 109°6′ E, Cai et al., 2010). (C) The Sanbao record (31°40′ N, 110°26′ E, Dong et al., 2010). (D) The Heshang record (30°27′ N, 110°25′ E, Hu et al., 2008). (E) The Hulu record (32°30′ N, 119°10′ E, Wang et al., 2001; Kelly et al., 2006). (F) The Lianhua record (29°29′ N, 109°33′ E, Cosford et al., 2008). (G) The Shigao record (28°11′ N, 107°10′ E, Jiang et al., 2012). (H) The Yamen record (25°29′ N, 107°54′ E, Yang et al., 2010). (I) The Dongge record (25°17′ N, 108°5′ E, blue: Wang et al., 2005; green: Dykoski et al., 2005). In panel (b), (A) Summer (JJA) mean Insolation at 65°N (Laskar et al., 2004). (B) The Tianmen record (30°55′ N, 90°4′ E, Cai et al., 2012). (C) The Timta record (29°50′ N, 80°2′ E, Sinha et al., 2005). (D) The Mawmluh record (25°16′ N, 91°43′ E, Berkelhammer et al., 2012). (E) The Hoti record (23°05′ N, 57°21′ E, Fleitmann et al., 2007). (F) The Qunf record (17°10′ N, 54°18′ E, Fleitmann et al., 2003). (G) The Moomi record (12°30′ N, 54° E, Shakun et al., 2007). Two vertical bars in (a) and (b) depict the Younger Dryas (YD) event and the Mystery Interval (including H1) (Denton et al., 2006), respectively. Cave locations are indicated in (c). Green stars show cave locations in the East Asian monsoon region: 1. Jiuxian cave, 2. Sanbao Cave, 3. Heshang cave, 4. Hulu cave, 5. Lianhua cave, 6. Shigao cave, 7. Yamen cave, 8. Dongge cave. Red stars show cave locations in the Indian monsoon region: 1. Tianmen cave, 2. Timta cave, 3. Mawmluh cave, 4. Hoti cave, 5. Qunf cave, 6. Moomi cave. The dashed line depicts the approximate fringe of the modern summer Asian monsoon.

[Figure]

**Fig. 2.** Time evolution of the last 20,000 years in observations and models. (a) Summer (JJA) insolation at 45°N (Berger, 1978), (b) $CO_2$ concentration from ice cores (Joos and Spahni, 2008), (c) meltwater flux in the Northern Hemisphere (black) and Southern Hemisphere (grey) in the TRACE simulation. (d) Greenland air temperature from GISP2 reconstruction (grey) and TRACE annual mean temperature (red), (e) $\delta^{18}O_c$ from Dongge and Hulu Caves (grey) (Wang et al., 2001), Chinese monsoon wind index (averaged summer meridional surface wind in East China (110°E–120°E; 27°N–37°N) in TRACE) (red), precipitation weighted annual Chinese $\delta^{18}O_p$ index (blue dot) and the $1\sigma$ spread (average domain (100°E–115°E, 27°N–36°N) in the snapshot simulations, and the model $\delta^{18}O_c$ corrected with annual (blue solid line) and JJA (blue circle) temperature ($\Delta T$) using the formula of $\Delta\delta^{18}O_c = \Delta\delta^{18}O_p - 0.24\text{‰ }°C^{-1}\Delta T$ (here $\Delta T$ is the anomaly from its temporal mean such that the correction only changes the amplitude of $\delta^{18}O_c$). The model $\delta^{18}O_s$ are shifted toward enrichment by $+2$‰ and the standard deviation is increased by a factor of 1.4. (f) summer rainfall in northern China (red) (110°E–120°E; 38°N–45°N) from TRACE and the lake level index from northern and southwestern China (grey), adapted from the late Fig. 5 as (1 × low lake level% + 2 × intermediate lake level% + 3 × high lake level%)/6, (g) summer rainfall in southeastern China (red) (110°E–120°E; 23°N–30°N) from TRACE and the pollen ratio of trees/shrubs from DaJiu lake (grey)) (Zhu et al., 2010).

conflicting results (LeGrande and Schmidt, 2009; Clemens et al., 2010; Pausata et al., 2011; Maher and Thompson, 2012; Tan, 2013). In particular, recent isotope modeling studies for the Holocene (LeGrande and Schmidt, 2009) and idealized Heinrich Event (Pausata et al., 2011) are consistent with the upstream depletion idea (Yuan et al., 2004; Hu et al., 2008) but contradict the idea that this depletion correlates with changes in local rainfall over China (Cheng et al., 2009). These isotope modeling studies concluded that the variability of the Chinese cave $\delta^{18}O_c$ records do not represent the variability of the EASM, but rather, rainfall change in the upstream source region, notably the Indian Ocean and the South Asian

monsoon region. Here, we propose a resolution that reconciles these seemingly conflicting ideas in light of the result of the first coupled general circulation model simulation of the evolution of the global climate and $\delta^{18}O$ over the last 21,000 years. Combining our simulation with paleo moisture records, and recognizing the characteristic spatial pattern of EASM rainfall response associated with the subtropical monsoon, we propose that the Chinese $\delta^{18}O$ records do represent the EASM, but monsoon intensity is best characterized by the southerly monsoon wind and accompanying rainfall in northern China. Furthermore, we suggest that the apparent correlation between depleted Chinese $\delta^{18}O$ and enhanced

rainfall in northern China is caused indirectly by the continental scale response of the Asian monsoon system to external climate forcing. The paper is arranged as follows. Section 2 will describe the models and experiments, and compare the model simulation with observations. Section 3 will study the relation between oxygen isotope and EASM from both the modeling and observational perspectives. Section 4 will further discuss the dynamics and robustness of the response of EASM in terms of monsoon winds and rainfall. Finally, in Section 5, we summarize the major conclusion and further discuss the potential mechanism that is responsible for the response of the Chinese $\delta^{18}O$ and monsoon rainfall from the broader perspective of the continental scale Asian monsoon system.

**2. Model and experiments**

We simulated continuous climate evolution of the last 21,000 years (Liu et al., 2009) in a state-of-art coupled ocean-atmosphere model, the Community Climate Model version 3 (CCSM3 T31 resolution) of the National Center for Atmospheric Research (hereafter TRACE simulation) (Yeager et al., 2003). The simulation is forced by realistic external forcing of insolation (Fig. 2a), atmospheric greenhouse gases (Fig. 2b), meltwater fluxes (Fig. 2c) and continental ice sheets (Liu et al., 2009; He, 2011). The TRACE simulation captures many major features of observed climate evolution, such as the global temperature (Shakun et al., 2012), the North Atlantic climate (Liu et al., 2009, 2012) and Southern Hemisphere climate (He, 2011).

To explicitly compare model results with $\delta^{18}O$ observations, we further simulated the evolution of atmospheric water isotopes using the isotope-enabled atmospheric component model of the CCSM3 CAM3 (T31 resolution) that incorporates fractionation associated with surface evaporation and cloud processes (Noone and Sturm, 2010). We performed 23 isotope snapshot sensitivity experiments in the last 21,000 years using the CAM3 set-up the same as in the TRACE experiment: 21 experiments are 1000 years apart, at 20 ka, 19 ka,…., 0 ka, and the additional 2 experiments are around the times of the Bølling−Allerød warming (BA) (14.5 ka) and Younger Dryas (YD) (12.1 ka). Each experiment is forced by the same external forcing as for the TRACE experiment, and additionally, by a 50-year history of monthly SST and sea ice from the TRACE experiment. Surface ocean $\delta^{18}O$ values are prescribed as $\delta^{18}O = 1.6‰$ at LGM based on (Schrag et al., 1996), and is reduced as an extrapolation following the sea level changes onto other periods, finally to 0.5‰ at 6 ka. The mean of the last 40 years are used for cross-snapshot analysis. The mean climate of each snapshot is similar to that of the TRACE simulation at the corresponding time. The prescribed ocean surface $\delta^{18}O$ boundary condition should not introduce large error because of the modest variation of the $\delta^{18}O$ value over the surface ocean. Therefore, the oxygen isotope simulated in the snapshot experiment can be considered a good representation of that simulated in the transient TRACE experiment, were the isotopes fully implemented into the coupled model.

The evolution of the model $\delta^{18}O_p$ over China captures the major features of the cave $\delta^{18}O_c$ records, albeit with a somewhat smaller magnitude (even after the correction of the cave temperature effect). This can be seen in the evolution of the Chinese $\delta^{18}O_p$ index, defined here as the annual mean $\delta^{18}O_p$ (weighted by the monthly precipitation) averaged over China, in comparison with the cave $\delta^{18}O_c$ record (Fig. 2e) as well as the model cave $\delta^{18}O_c$ that is derived after the correction of the cave annual temperature change ($\Delta T$) using the equation $\Delta\delta^{18}O_c = \Delta\delta^{18}O_p - 0.24‰ \, °C^{-1} \, \Delta T$, as in Pausata et al. (2011). Both the model $\delta^{18}O_p$ (blue dots) and $\delta^{18}O_c$ (blue solid line) follow the overall evolution of the observation (grey line) $\delta^{18}O_c$: all are enriched during the Heinrich Event 1 (H1, ~17ka),

depleted during the Bølling−Allerød warming (BA, ~14.6 ka), enriched modestly during the Younger Dryas (YD, ~12 ka), and trend toward more enriched values through most of the Holocene (Fig. 2e). The temperature effect increases the variability amplitude of model $\delta^{18}O_c$ over model $\delta^{18}O_p$ by ~40% for annual temperature, (and by another 30% if the summer temperature is used (Fig. 2e)), but is still weaker than the observation (note in Fig. 2e, the scale of the model $\delta^{18}O$ is increased by a factor of 1.4). Furthermore, consistent with the coherent $\delta^{18}O_c$ variability across different caves in China (Fig. 1a), the model $\delta^{18}O_p$ variability is also coherent over almost all of eastern China, as shown in the cross-snapshot correlation map of the annual $\delta^{18}O_p$ with the Chinese $\delta^{18}O_p$ index (Fig. 3a). Finally, consistent with previous isotope modeling studies (LeGrande and Schmidt, 2009; Pausata et al., 2011), and the coherent cave records across the East (Fig. 1a) and South (Fig. 1b) Asia, the evolution of the $\delta^{18}O_p$ over China is highly correlated with $\delta^{18}O_p$ along the moisture transport route upstream in tropical Indian Ocean, South and Southeast Asia monsoon (Fig. 3a), rather than the local rainfall in southeastern China. This lack of local correlation can be seen more clearly in the local correlation map between $\delta^{18}O_p$ and rainfall (Fig. 4), which shows little correlation in eastern China. This is also consistent with the upstream depletion mechanism originally proposed by Yuan et al. (2004) and Hu et al. (2008).

**3. Oxygen isotope and East Asian monsoon climate**

We now show that the coherent $\delta^{18}O_p$ signal over China, although caused significantly by rainfall change in upstream, is still able to represent key features of the EASM, notably the monsoon wind and the monsoon rainfall in northern China. This can be seen first in the cross-snapshot correlation map between the Chinese $\delta^{18}O_p$ index and monsoon rainfall and low level winds (Fig. 3b). The Chinese $\delta^{18}O_p$ shows little correlation with rainfall in southeastern China, but shows significant correlation with monsoon rainfall in northern China, with the high correlation region stretching into southwestern China along the margin of EASM. Most notably, the Chinese $\delta^{18}O_p$ is highly correlated with southerly monsoon winds over eastern China (Fig. 3b), with the cross-snapshot correlation coefficient between the Chinese $\delta^{18}O$ index and the 850 hpa meridional wind higher than 0.7 and 0.8 over almost the entire eastern China (not shown). As such, depleted $\delta^{18}O_p$ corresponds to intensified southerly monsoon winds and increased rainfall in northern China, but little rainfall variability in southeastern China. The similarity of the $\delta^{18}O_p$ and monsoon wind can also be seen in the resemblance of cross-snapshot correlation maps of the $\delta^{18}O_p$ distribution with the $\delta^{18}O_p$ index (Fig. 3a) and with the monsoon wind index (Fig. 3c), the latter being defined as the summer southerly wind averaged over eastern China. Indeed, the correlation maps between the monsoon rainfall and the wind and $\delta^{18}O_p$ indices (Fig. 3b and d) suggest that the millennial response of the EASM can be viewed as that of a part of the grand Asian−African monsoon system, with the rainbelt stretching from northern China to southwestern China and Southeast Asia, further westward across the Indian subcontinent and the Arabia peninsula, and eventually into the North Africa (Fig. 3b). This continental scale response pattern is related to the mechanism of the millennial monsoon response to global climate forcing. We will return to this point later for a further understanding of the $\delta^{18}O_p$.

The correspondence between the EASM monsoon winds and the $\delta^{18}O_p$ values is further seen in the consistent evolution between the isotopes (in the model and cave) and the monsoon wind index in TRACE (Fig. 2e). Overall, monsoon winds and $\delta^{18}O_p$ depletion both increased from the LGM toward the early Holocene and then decreased in the Holocene, following the summer

**(a) <-d18O_ind, d18O>**

**(c) <V_ind, d18O>**

**(b) <-d18O_ind, Prec>**

**(d) <V_ind, Prec>**

**Fig. 3.** Cross-snapshot correlation between the (negative) Chinese $\delta^{18}O_p$ index and (a) the annual $\delta^{18}O_p$ (weighted by monthly precipitation) and (b) summer (JJA) rainfall. The vectors are (a) the climatological total column moisture transport for the summer, and (b) the regression coefficient of the 850 hPa wind on the Chinese $\delta^{18}O_p$ index. (c) and (d) are the same as (a) and (b), respectively, except that the index is replaced by the Chinese monsoon wind index, which is calculated as the summer southerly wind averaged in eastern China (110E−120E; 27N−37N).

insolation. Both the winds and the $\delta^{18}O_p$ are punctuated by millennial-scale events associated with meltwater pulses. These evolutionary features of the Chinese monsoon winds and $\delta^{18}O_p$ depletion resemble those of the Greenland temperature (Fig. 2d),

albeit with less pronounced millennial-scale events during the deglacial.

In contrast to monsoon winds, the monsoon rainfall exhibits complex regional differences in eastern China. The monsoon

[Figure]

**Fig. 4.** Point-to-point cross-snapshot correlation map between summer (JJA) rainfall and monthly precipitation weighted $\delta^{18}O_p$ among the snapshot experiments. (90% SigLevel = 0.4). The insignificant correlation in the East China suggests the lack of correlation between the $\delta^{18}O_p$ and local rainfall there.

[Figure]

[Figure]

**Fig. 5.** (a) Lake level changes since 22 ka in the East China (95–132°E): a histogram showing the total 14 numbers of lakes of high, intermediate, and low levels through time. Data are taken from the Chinese Lake Status Database (Yu et al., 2001) with updated revision (Yu et al., 2013). (b) Lake locations: 1. Baisuhai Lake, 2. Chagangnur Lake, 3. Erjichuoer Lake, 4. Jilantai Salt Lake, 5. Salawusu Paleolake, 6. Xidadianzi Lake, 7. Xingkai Lake, 8. Baijan Lake, 9. Hulun Lake, 10. Fuxian Lake, 11. Manxing Lake, 12. Ningjinbo Lake, 13. Shayema Lake, 14. Erhai Lake. The variability remains similar if the lakes are analyzed in two subgroups in northern China and in southwestern China. To facilitate with the comparison of model rainfall, these three levels are weighted averaged to give a lake moisture index in Fig. 1f. In contrast to the extensive lake level records in the northern and southwest China, there are few reliable lake level records in southeast China. Lakes in the southeast China are mostly overflowing and the sediments are seasonally influenced by alluvial processes of both erosions and deposition (Yu et al., 2001). These features may lead the lacustrine sediments to discontinuous sequence and, in turn, poor preservation of the lake level records.

rainfall in northern China (red line, Fig. 2f) intensifies toward the early Holocene and weakens gradually afterwards (red line, Fig. 2f), largely resembling the $\delta^{18}O_p$ depletion (Fig. 2e), both reminiscent of orbital forcing and deglacial meltwater forcing. In contrast, monsoon rainfall in southeastern China increases from the LGM all the way to late Holocene (red line, Fig. 2g), with little resemblance to any single forcing. These simulated regional rainfall responses are largely consistent with the traditional paleoclimate records for moisture, such as lake status and pollen records. In northern China, the model rainfall is consistent with the extensive lake level records

there (grey line, Figs. 2f and 5) (Yu et al., 2013), both largely following the summer insolation. In southeastern China where traditional moisture proxies are extremely limited, a recent lake sediment record shows an increased pollen ratio between tree and shrub, implying increased rainfall from the deglacial period to the late Holocene (grey line, Fig. 2g), also consistent with the model. Indeed, the overall evolution feature of the model rainfall during the Holocene is consistent with a synthesis of the traditional moisture proxies over China, which shows an optimum moisture climate in the early Holocene in northern China, but in the

late Holocene in southeastern China (An et al., 2000; Wang et al., 2010).

The different rainfall responses between northern and southern China and the associated monsoon winds have also been found robust in observations across a broad range of time scales, such as the present climate variability on interannual to interdecadal time scales (Ding et al., 2007) and the trend of the last 50 years (Fig. 6). The different rainfall responses between northern and southern China has also been found for centennial variability of the last millennia in historical records (Wang et al., 1987; Ge et al., 2013).

The characteristic EASM rainfall response pattern is also dominant at millennial to orbital time scales, as seen in the two leading Empirical Orthogonal Functions (EOFs) of the summer rainfall in the TRACE simulation (Fig. 7) (Similar for annual rainfall, not shown). EOF1 (Fig. 7a) is dominated by the Asian−African monsoon rainbelt, similar to that associated with Chinese $\delta^{18}O_p$ (Fig. 3a) or monsoon winds (Fig. 3c). In eastern China, significant increase of rainfall in northern China is accompanied by enhanced southerly winds, with little or opposite rainfall response in southeastern China. The pattern evolution (PC1 in Fig. 7a) also resembles the $\delta^{18}O$ or monsoon wind indices (Fig. 2e). This evolution is similar to the PC1 or global mean of the surface temperature in the reconstruction and the TRACE simulation (Shakun et al., 2012) and is forced predominantly by the greenhouse gases and orbital forcing. The different rainfall responses between northern and southern China can also be seen in the EOF2 of rainfall (Fig. 7b), which, as seen in its temporal variability in PC2, is forced predominantly by the meltwater through Atlantic Meridional Overturning Circulation and the interhemisphere temperature gradient (Shakun et al., 2012).

**4. The responses of the EASM wind and rainfall**

The characteristic pattern of the EASM rainfall/wind discussed above is determined by the dynamics of the EASM, a subtropical monsoon that differs from a tropical monsoon like the Indian Summer Monsoon (Wang and Lin, 2002; Ding and Chan, 2005). As a subtropical monsoon, the EASM is associated closely with the low level southerly wind, and the migration of the rainbelt known as the Meiyu Front in China, which lies on the northwestern flank of the North Pacific Subtropical High. As the Meiyu Front/Subtropical High system migrates, rainfall varies significantly from southern to

northern China. As such, it is difficult to represent the entire EASM system with a single index (Wang et al., 2008), especially in regional precipitation. The southerly wind has been used traditionally as an index for the EASM with clear dynamic implications on moisture import (Wu and Ni, 1991; Ding and Chan, 2005). An intensified low level southerly wind, which is associated with a stronger western Pacific Subtropical High, enhances the moisture transport and the leading moisture front deep into northern China, where the front interacts with the synoptic weather systems coming down from the mid-and-high latitudes, leading to more frequent and heavier rainfalls there.

We further show that the above discussed responses of EASM monsoon winds and rainfall are also robust across a broad range of climate models. First, it is known that models generally simulate the wind field much better than the precipitation field, because the latter depends on the difference (gradient) of the former and therefore is subject to greater model errors. Therefore, it is reasonable to expect a more consistent EASM response across models in monsoon winds than in rainfall. We first examine model responses to orbital forcing. From late to middle Holocene, our CCSM3 simulates a greatly enhanced EASM, which is characterized by enhanced southerly winds in eastern China, an increased rainfall in northern China, and weakly reduced rainfall in southern China (Fig. 8). The enhanced southerly winds and monsoon rainfall in northern China in summer is consistent with the simulation in the GISS model (LeGrande and Schmidt, 2009) (although the annual mean rainfall response in China in the latter is small and less significant). The $\delta^{18}O$, however, is depleted over entire eastern China (Fig. 8), as in GISS model. Indeed, many previous models have simulated different regional rainfall responses in eastern China (e.g. An et al., 2000; Liu et al., 2003; Shi et al., 2012). The regional responses seem to be forced predominantly by the direct insolation forcing over land, as shown in some model studies where the SST is fixed (Liu et al., 2003). To further assess the robustness of the model response to orbital forcing systematically, we examine the responses in an ensemble of 10 state-of-art models (PIMP3) for the mid-Holocene (Zheng et al., 2012). The ensemble mean summer response shows an enhanced southerly wind and the accompanying rainfall in northern China, and both features are highly consistent across models, with the ensemble mean exceeding the ensemble spread (Fig. 9a and b). In comparison, the rainfall in southeastern China shows little consistence among models, with the ensemble mean smaller than the ensemble spread (Fig. 9a). This summer ensemble rainfall response is also largely consistent in the annual rainfall (Fig. 9c).

We now examine the model responses to meltwater forcing. For the convenience of discussing an enhanced EASM and depleted $\delta^{18}O$, we will discuss the negative sign of the hosing experiment, with the anomaly derived as the control experiment minus hosing experiment, equivalent to the effect of the termination of a hosing or simply called "de-hosing" here. We first show an idealized de-hosing experiment in our CCSM3. It is seen that, with the termination of the meltwater, Summer Monsoon winds are increased across eastern China accompanied by an intensified rainfall in northern China ($>35°N$) and a reduced rainfall over most of the southeastern China (Fig. 10) This rainfall response (and the depleted $\delta^{18}O$) is consistent with the GISS model (Lewis et al., 2010). The intensified southerly winds are consistent with a GFDL model study (Zhang and Delworth, 2005). However, the summer rainfall responses in the GFDL model increases over the entire eastern China, except in between 25oN and 30°N where the rainfall response is weak. These rainfall responses are different from yet another model (Pausata et al., 2011), which showed little rainfall response across the entire eastern China. For a more systematic examination of the climate response to meltwater, we examine the

[Figure]

**Fig. 6.** Linear trend of annual rainfall from 1957 to 2007 (in %) in 170 stations across China. The rainfall trend exhibits a significant decrease from northern China to southwest China, but opposite and insignificant rainfall changes in southeastern China. Accompanied with this monsoon rainfall reduction in northern China is a significant reduction of the southerly monsoon wind (Ding et al., 2007).

[Figure]

**Fig. 7.** The first (a) (upper panels) and second (b) (lower panels) EOF of the summer (JJA) rainfall over global land in the transient simulation. The left panels show the EOF pattern and the right panels show the principle components. The vectors are the 850 hpa wind regressed on the principle components. These leading EOFs and principle components remain similar if the domain is limited to the Asian region, or East Asian region. Of particular note here is the different rainfall variability in northern and southeast China, and the associated wind field toward the region of increased rainfall. This feature remains similar if the EOF domain is reduced to the Asia or East Asia only, and for the annual rainfall. The patterns of the two rainfall EOFs resemble the two leading EOFs of the observed interdecadal rainfall in China in the last 50 years (Ding et al., 2007).

hosing experiments in an ensemble of 13 models, including some of our modified versions of the CCSM3 (Fig. 11). The ensemble mean Summer Monsoon winds are increased in eastern China, and, in particular, are consistent along the east coast (Fig. 11b). The rainfall responses in the summer (Fig. 11a) and annual mean (Fig. 11c) show an intensification over most of China, but most consistent in northern China and least consistent in southeastern China (Fig. 11a). These features of model consistency are similar to the response to orbital forcing discussed earlier (Fig. 9). Therefore, the robust responses of monsoon wind and northern China monsoon rainfall, and the least robust response in southwestern China rainfall, are all robust features across models. All these evidences support the notion that the wind response is more robust than the rainfall response for the EASM and therefore is a better variable for representing the EASM.

**5. Discussion and conclusions**

Our study above leads us to conclude that, for millennial climate changes, the Chinese cave $\delta^{18}O_c$ record is a robust indicator of the EASM in terms of the monsoon wind and the accompanying rainfall in northern China, even though the change of the Chinese cave $\delta^{18}O_c$ is determined significantly by upstream depletion. This

conclusion clarifies the interpretation of the cave records on the EASM (Wang et al., 2001; Yuan et al., 2004; Hu et al., 2008; Cheng et al., 2009) in a consistent dynamic framework. One key element that leads us to this conclusion is the recognition that, as a sub-tropical monsoon, the monsoon rainfall of EASM exhibits a characteristic response pattern within China. This conclusion, we believe, is robust. First, our interpretation of the characteristic pattern of the EASM monsoon winds and rainfall is consistent with the dynamics of EASM. A stronger southerly enhances the moisture transport and advances the monsoon front to northern China, where the fronts interact with weather systems from the mid-and-high latitude to produce more rainfall there. Second, the different rainfall responses in northern and southern China are consistent with observations of the past and present climate variability across a broad range of time scales. Third, our isotope-climate simulation in CCSM3 is consistent with the two currently published isotope modeling studies, one for the Holocene (LeGrande and Schmidt, 2009) and the other for an idealized H1 (Pausata et al., 2011). These isotope modeling studies are also consistent with the coherent cave records across the Asia monsoon region (Fig. 1).

We now further discuss the isotope and EASM in the broader context of the Asian monsoon, with a special attention to the relationship between EASM and South Asian monsoon.

[Figure]

**Fig. 8.** The difference between middle (6 ka) and late (1 ka) Holocene for (a) JJA precipitation and 850 hpa wind, (b) annual total precipitation, (c) JJA δ$^{18}$O (precipitation weighted), (d) annual δ$^{18}$O (precipitation weighted). The response is forced mainly by orbital forcing. In eastern China, the monsoon wind is intensified, accompanied by increased rainfall in northern China and reduced rainfall in southern China in both the summer and annual total. The δ$^{18}$O, however, is depleted over entire eastern China.

**1) Moisture sources**

The upstream depletion for the EASM may be related to its nature of a subtropical monsoon. Indeed, although poorly correlated with local rainfall for the subtropical monsoon over East China, δ$^{18}$O$_p$ correlates well with local rainfall over much of the tropical monsoon region, such as India and Southeast Asia (Fig. 4). This difference in correlation between δ$^{18}$O$_p$ and local rainfall may reflect, partly, the different nature of the subtropical monsoon and tropical monsoon. Tropical monsoon rainfall occurs nearby its moisture source with a large rainfall amount (relative to that in its source region), such that the δ$^{18}$O$_p$ is affected more by local processes. In contrast, subtropical monsoon rainfall tends to occur far away from its moisture source with only a modest amount of rainfall, rendering the δ$^{18}$O$_p$ affected more by upstream depletion.

As a preliminary study of the remote moisture sources of the EASM, we back-track air parcels for the present day world using the NOAA backward trajectory model HYSPLITv4.9 (Draxler and Hess, 1998) and the NCEP reanalysis meteorological fields. The remote moisture source of the EASM for the summer (JJA) of 1982—2011 is back-tracked for air parcels from the Yangtze-Huaihe river valley (110—121°E, 28—34°N, red line box marked in Fig. 12a) (Jiang et al., 2013), a typical EASM sub-region in southeastern China that includes the site of the Hulu cave. Every 6 h, 24 parcels are started at the level of 850 hPa, roughly each in a 2° × 2° grid box. All the

trajectories are integrated backward using the 6 hourly NCEP reanalysis field for 11 days, when the average trajectory positions have reached quasi-equilibrium. The density of final parcel positions is then calculated as the number of parcels in each 2° × 2° grid box (Fig. 12a, shading). It is seen that there are three major air mass sources. The dominant air mass sources is from the southwestern source over the Indian Ocean via the South China Sea (48%, 30°E—120°E, 0—22°N) following the southwesterly monsoon wind; the second source is the southeastern source from the western North Pacific (29%, 120°E—200°E, 0—50°N) following the trade wind on the southern flank flow of the western North Pacific Subtropical High; finally, there is a third northern source from the broad high latitude region across the Europe to Siberia (18%, 22—80°N, outside the other source regions) following the mid-level westerly wind (the remaining ~5% locally over eastern China (110—120°E, 22—45°N)). The specific humidity (Fig. 12a, contour), temperature and height map for these final positions (Fig. 12b) further show that the southwestern source (from the Indian Ocean) and, to some extent, the southeastern source (western Pacific) consist of warm surface air with high moisture content, while the northern branch consists of cold mid-atmosphere dry air. Therefore, the contribution of moisture source, which can be estimated as the parcel number multiplying the specific humidity, is contributed overwhelmingly by the southwestern source (59%) and, then, by the southeastern source (30%). Therefore, most of the moisture source of the EASM

**a) Prep %, JJA**

[Figure]

**b) V 850hpa, JJA**

**c) Prep %, Annual Mean**

**Fig. 9.** The changes of (a) summer precipitation (%) and (b) 850 hpa meridional wind (m s$^{-1}$) of JJA and (c) annual mean precipitation (%) in response to orbital forcing in the mid-Holocene (6 ka—0 ka) for the ensemble mean of the PMIP3 models with equal weight. The shading indicates regions where the multi-model ensemble mean exceeds the inter-model standard deviation, or the region of more consistent or robust model responses. The PMIP3 models are identical to those used in Zheng et al. (2012) (BCC-CSM1-1, CCSM4, CNRM-CM5, CSIRO-Mk3-6-0, FGOALS-g2, FGOALS-s2, FIO-ESM, IPSL-CM5A-LR, MPI-ESM-P, MRI-CGCM3). It is seen that the summer monsoon wind is increased consistently in eastern China across models; the rainfall tends to increase increases tend to be consistent in northern to southwestern China, but changes less and inconsistent in southeastern China.

originates predominantly from the remote Indian Ocean region. This result seems to be determined by the large scale monsoon circulation field and is consistent with the high correlation of precipitation $\delta^{18}O_p$ between eastern China and the upstream Indian Ocean region simulated in the model (Fig. 3a).

2) Dynamics of the continental scale monsoon response

Now, we further attempt to reconcile the two seemingly contradictory ideas of the Chinese cave $\delta^{18}O$, which, on the one hand, is determined by the upstream depletion, on the other

[Figure]

**Fig. 10.** The response of summer (JJA) (top) winds on 850 hpa (m/s) and precipitation (mm/day) and (bottom) sea level pressure (contour, pa) and surface air temperature (shading, °C) to the termination of North Atlantic hosing (1 Sv. of 100 years) in CCSM3 (T31 version) at the glacial state (i.e. LGM − LGM hosing). With the activation of the Atlantic Meridional Overturning Circulation, the southwesterly monsoon wind is enhanced, accompanied by an increase of rainfall in northern China.

hand, yet is a robust indicator of the EASM intensity in terms of the monsoon wind and rainfall in northern China. In spite of the upstream depletion, the millennial change of the Chinese $\delta^{18}O$ depletion can still appear correlated with the rainfall in northern China. This follows because the grand Asian–African monsoon system responds to external global climate forcing in a coherent continental scale response that is characterized by a monsoon rainbelt, which consists of the response of the EASM monsoon wind and rainfall locally in northern China, the South and Southeast Asia monsoon remotely in the upstream region (Fig. 1a and b) and even the northern Africa monsoon far remote from the East Asia (e.g. Kutzbach and Street-Perrott, 1985; Liu et al., 2003; Chen et al., 2010; Fig. 7). Therefore, the apparent correlation between the $\delta^{18}O_p$ depletion in China and the rainfall increase in northern China is caused indirectly by the coherent monsoon rainfall responses in northern China and the upstream regions, rather than directly by the local rainfall process.

Physically, this coherent continental scale monsoon response to orbital and meltwater forcing can be understood as follows. First, with an increased insolation in summer (as of early or mid- Holocene relative to late Holocene, e.g. Figs. 8 and 9), surface temperature increases more over land than ocean, leading to an increased land–sea temperature contrast. In the tropical monsoon region (upstream of the EASM) such as the India and Southeast Asia, the increased land–sea temperature contrast induces monsoon low surface pressure over land, increasing the lower

layer convergence and in turn the monsoon rainfall there. In the subtropical monsoon region of the East Asia, the increased land–sea temperature contrast enhances the western North Pacific Subtropical High. The increased Subtropical High is accompanied by a stronger southerly wind over East China on its western flank, which increases moisture transport to, and therefore the monsoon rainfall in, northern China. Second, in response to a termination of the meltwater in the North Atlantic (the negative sign of the hosing response, e.g. Figs. 10 and 11), the northward migration of the ITCZ that is associated with an enhanced Atlantic Meridional Overturning Circulation leads to a warming over the tropical North Atlantic and in turn tropical Indian Ocean through the atmospheric Kelvin wave (Dong and Sutton, 2002). Thermodynamically, the warmer Indian Ocean increases the rainfall over the ocean and the Indian monsoon through a greater surface moisture supply. In the meantime, dynamically, the warmer Indian Ocean excites an atmospheric Kelvin wave to the western Pacific and intensifies the western North Pacific Subtropical High (Xie et al., 2009), which then increases the southerly wind in East China and, in turn, moisture supply and rainfall in northern China. Finally, in response to either orbital or meltwater forcing, the response of the South Asian monsoon can also influence northern China through the excitation of the so called "Silk Road" atmospheric teleconnection (Enomoto et al., 2003; Chen et al., 2010). Indeed, current observations also show that the interannual variability of the monsoon rainfall in Indian is positively correlated with that in China most significantly in northern China (Guo and Wang, 1998; Lian, 1998).

[Figure]

[Figure]

[Figure]

**Fig. 11.** The responses of summer (JJA) (a) precipitation (%) and (b) 850 hpa meridional wind (m s$^{-1}$) and (c) annual mean precipitation (%) for the ensemble mean of 13 models (equal weights) in response to the termination of North Atlantic hosing (CTRL-Hosing). The shading indicates the regions where the ensemble mean exceeds the ensemble standard deviation. The models are: CCSM3_T42 at LGM (1 Sv. hosing), CCSM3_T31 at LGM (1 Sv. hosing), CCSM3_T31 at Mid-Holocene (0.38 Sv hosing), FGOALS-g2 at pre-industry (0.38 Sv. hosing), GFDL_CM2 at present (0.6 Sv. hosing, Courtesy to Rong Zhang), and 8 models at the present using 1 Sv. hosing: CCSM3_T31, CCSM3_ADJ (glob1e heat-freshwater flux adjustment applied), CCSM3_DPL (a dipole of freshwater flux adjustment applied over the tropical Atlantic), GFDL_CM2.1, GFDL_R30, HadCM3, CCSM2_T42, Univ. Toronto (last 5 models, courtesy to J.J. Yin, R. Stouffer and S. Q. Zhang). All 13 experiments are used for the annual mean response of precipitation in (c) while only the first 8 models are available for the summer precipitation and 850 hPa meridional wind in (a) and (b). Overall, the southwesterly wind is enhanced consistently in eastern China; the rainfall tends to be increased consistently in northern China, but not in southwestern China.

**Fig. 12.** Spatial distribution of back-trajectory calculation of air parcels −11 days from the Yangtze-Huaihe valley (red box in (a)) at the level of 850 hPa. (a) particle density (particle number/2° × 2° grid box, shading) and specific humidity (g/kg, contour), (b) temperature (°C, shading) and height (hPa, contour). In (a), the three arrows represent schematically the three major air sources from the southwest (Indian Ocean and South China Sea), southeast (western North Pacific) and northern high latitudes; the numbers for each arrow indicates the percentage of particles from each source; the numbers within the parentheses indicate the percentage of moisture from each source. The back-tracking was done with 24 parcels released every 6 h and back-tracked for 11 days using the 6 hourly NCEP reanalysis of 1982−2011.

This positive correlation is caused by atmospheric teleconnections (Kriplani and Kulkarni, 2001), consistent with our model results. It should be noted, however, that this coherent continental scale variability of the Asian monsoon does not seem to be related to the rainfall change in southeastern China. It is possible that rainfall in southeastern China is controlled by more complex dynamics and depends on, in addition to moisture supply, the monsoon front, tropical depressions and other factors (Ding et al., 2009). The

dynamics of the EASM rainfall in southeastern China deserves much further studies in the future.

**Acknowledgments**

The authors would like to thank the helpful discussions with Drs. D. Battisti, Ming Tan, Shaowu Wang and Zhishen An. This research is funded by NSFC41130105, CAS/SAFEA International Project (KZZD-EW-TZ-08), NSFC41230524, NSFC40921120406 and US NSF and DOE. The computation is carried out at Oak Ridge National Lab and the NCAR supercomputing facility. This paper is CCR contribution 1162.

**References**

An, Z., Porter, S.C., Kutzbach, J.E., Wu, X., Wang, S., Liu, X., Li, X., Zhou, W., 2000. Asynchronous Holocene optimum of the East Asian monsoon. Quat. Sci. Rev. 19, 743–762.

Berger, A., 1978. Long-term variations of daily insolation and Quaternary climatic changes. J. Atmos. Sci. 35, 2362–2367.

Berkelhammer, M., Sinha, A., Stott, L., Cheng, H., Pausata, F., Yoshimura, K., 2012. An abrupt shift in the Indian Monsoon 4,000 years ago. In: AGU Geophysical Monograph on Climate and Civilization. http://dx.doi.org/10.1029/2012GM00120.

Cai, Y.J., Tan, L.C., Cheng, H., An, Z.S., Edwards, R.L., Kelly, M.J., Kong, X.G., Wang, X.F., 2010. The variation of summer monsoon precipitation in central China since the last deglaciation. Earth Planet. Sci. Lett. 291, 21–31.

Cai, Y.J., Zhang, H.W., Cheng, H., An, Z.S., Edwards, R.L., Wang, X.F., Tan, L.C., Liang, F.Y., Wang, J., 2012. The Holocene Indian monsoon variability over the southern Tibetan Plateau and its teleconnections. Earth Planet. Sci. Lett. 335, 135–144.

Chen, G.S., Liu, Z., Clemens, S., Prell, W., Liu, X., 2010. Modeling the time-dependent response of the Asian summer monsoon to obliquity forcing in a coupled GCM: a PHASEMAP sensitivity experiment. Clim. Dyn. 36, 695–710. http://dx.doi.org/10.1007/s00382-010-0740-3.

Cheng, H., Edwards, L., Broecker, W., Denton, G., Kong, X., Wang, Y., Zhang, R., Wang, X., 2009. Ice age terminations. Science 326, 248–252.

Cheng, H., Sinha, A., Wang, X., Cruz, F., Edwards, L., 2012. The Global Paleomonsoon as seen through speleothem records from Asia and the Americas. Clim. Dyn. 39, 1045–1062. http://dx.doi.org/10.1007/s00382-012-1363-7.

Clemens, S., Prell, W., Sun, Y., 2010. Orbital-scale timing and mechanisms driving late Plesitocene Indo-Aisan summer monsoons: reinterpreting cave speleothem δ¹⁸O. Paleoceanography 25, PA4207. http://dx.doi.org/10.1029/2010PA001926.

Cosford, J., Qing, H.R., Eglington, B., Mattey, D., Yuan, D.X., Zhang, M.L., Cheng, H., 2008. East Asian monsoon variability since the Mid-Holocene recorded in a high-resolution, absolute-dated aragonite speleothem from eastern China. Earth Planet. Sci. Lett. 275, 296–307.

Denton, G.H., Broecker, W.S., Alley, R.B., 2006. The mystery interval 17.5 to 14.5 kyrs ago. Pages News 14, 14.

Ding, Y.H., Chan, C.L., 2005. The East Asian summer monsoon: an overview. Meteorol. Atmos. Phys. 89, 117–142.

Ding, Y., Wang, Z., Sun, Y., 2007. Interdecadal variation of the summer precipitation in East China and its association with decreasing Asian monsoon. Part I: observed evidences. Int. J. Climatol. 28, 1139–1161. http://dx.doi.org/10.1002/joc.1615.

Ding, Y., Sun, Y., Wang, Z., Zhu, Y., Song, Y., 2009. Inter-decadal variation of the summer precipitation in China and its association with decreasing Asian summer monsoon. Part II: Possible causes. Int. J. Climatol 29, 1926–1944. http://dx.doi.org/10.1002/joc.1759.

Dong, B.W., Sutton, R.T., 2002. Adjustment of the coupled ocean–atmosphere system to a sudden change in the thermohaline circulation. Geophys. Res. Lett. 29, 1728. http://dx.doi.org/10.1029/2002GL015229.

Dong, J.G., Wang, Y.J., Cheng, H., Hardt, B., Edwards, R.L., Kong, X.G., Wu, J.Y., Chen, S.T., Liu, D.B., Jiang, X.Y., Zhao, K., 2010. A high-resolution stalagmite record of the Holocene East Asian Monsoon from Mount Shennongjia, central China. Holocene 20, 257–264.

Draxler, R.R., Hess, G.D., 1998. An overview of HYSPLIT_4 modeling system for trajectories dispersion and deposition. Aust. Meteorol. Mag. 47, 295–308.

Dykoski, C.A., Edwards, R.L., Cheng, H., Yuan, D.X., Cai, Y.J., Zhang, M.L., Lin, Y.S., Qing, J.M., An, Z.S., Revenaugh, J., 2005. A high-resolution, absolute-dated Holocene and deglacial Asian monsoon records from Dongge cave, China. Earth Planet. Sci. Lett. 233, 71–86.

Enomoto, T., Hoskins, B.J., Matsuda, Y., 2003. The formation mechanism of the Bonin high in August. Q. J. R. Meteorol. Soc. 129, 157–178.

Fleitmann, D., Burns, S.J., Mudelsee, M., Neff, U., Kramers, J., Mangini, A., Matter, A., 2003. Holocene forcing of the Indian monsoon recorded on a stalagmite from southern Oman. Science 300, 1737–1739.

Fleitmann, D., Burns, S.J., Mangini, A., Mudelsee, M., Kramers, J., Villa, I., Neff, U., Al-Subbary, A.A., Buettner, A., Hippler, D., Matter, A., 2007. Holocene ITCZ and Indian monsoon dynamics recorded in stalagmites from Oman and Yemen (Socotra). Quat. Sci. Rev. 26, 170–188.

Gat, J.R., 1996. Oxygen and hydrogen isotopes in the hydrological cycle. Annu. Rev. Earth Planet. Sci. 24, 225–262.

Ge, Q.S., Zhang, J.Y., Hao, Z.X., Liu, H.L., 2013. General characteristics of climate changes during the past 2000 years in China. Sci. China, Earth Sci. 56, 321–329. http://dx.doi.org/10.1007/s11430-012-4370-y.

Guo, Q., Wang, J., 1998. A comparison of the summer precipitation in India with that in China. J. Trop. Meteorol. 4, 53–60 (in Chinese).

He, F., 2011. Simulating Transient Climate Evolution of the Last Deglaciation with CCSM3. Ph.D thesis. Department of Atmospheric and Oceanic Sciences, University of Wisconsin-Madison, p. 161.

Hu, C.Y., Henderson, G.M., Huang, J.H., Xie, S.C., Sun, Y., Johnson, K.R., 2008. Quantification of Holocene Asian monsoon rainfall from spatially separated cave records. Earth Planet. Sci. Lett. 226, 221–232.

Jiang, X.Y., He, Y.Q., Shen, C.-C., Kong, X.G., Li, Z.Z., Chang, Y.-W., 2012. Stalagmite inferred Holocene precipitation in northern Guizhou Province, China, and asynchronous termination of the Climatic Optimum in the Asian monsoon territory. Chin. Sci. Bull. 57, 795–801.

Jiang, Z., Ren, W., Liu, Z., Yang, H., 2013. Analysis of water vapor transport characteristics during Meiyu over the Yangtze-Huaihe River valley using the Lagrangian hethod. Acta Meteorol. Sin. 71 (2), 295–304.

Joos, F., Spahni, R., 2008. Rates of change in natural and anthropogenic radiative forcing over the past 20,000 years. Proc. Natl. Acad. Sci. U.S.A. 105, 1425.

Kelly, M.J., Edwards, R.L., Cheng, H., Yuan, D.X., Cai, Y., Zhang, M.L., Lin, Y.S., An, Z.S., 2006. High resolution characterization of the Asian Monsoon between 146,000 and 99,000 years B.P. from Dongge Cave, China. Paleogeogr. Paleoclimatol. Paleoecol. 236, 20–38.

Kriplani, R.H., Kulkarni, A., 2001. Monsoon rainfall variations and teleconnections over south and east Asia. Int. J. Climatol. 21, 603–616.

Kutzbach, J., Street-Perrott, F., 1985. Milankovitch forcing of fluctuations in the level of tropical lakes from 18 to 0 kyr BP. Nature 317, 130–134.

Laskar, J., Robutel, P., Joutel, F., Gastineau, M., Correia, A.C.M., Levrard, B., 2004. A long term numerical solution for the insolation quantities of the Earth. Astron. Astrophys. 428, 261–285.

LeGrande, A., Schmidt, G., 2009. Sources of Holocene variability of oxygen isotopes in paleoclimate archives. Clim. Past 5, 441–455.

Lewis, S.C., LeGrande, A.N., Keller, M., Schmidt, G.A., 2010. Water vapour source impacts on oxygen isotope variability in tropical precipitation during Heinrich events. Clim. Past 6, 325–343.

Lian, P.D., 1998. Indian summer monsoon and rainfall in North China. Acta Meteorol. Sin. 46, 75–81 (in Chinese).

Liu, Z., Otto-Bliesner, B., Kutzbach, J., Li, L., Shields, C., 2003. Coupled climate simulations of the evolution of global monsoons in the Holocene. J. Clim. 16, 2472–2490.

Liu, Z., Otto-Bliesner, B., He, F., Brady, E., Clark, P., Lynch-Steiglitz, J., Carlson, A., Curry, W., Brook, E., Jacob, R., Erickson, D., Kutzbach, J., Cheng, J., 2009. Transient simulation of deglacial climate evolution with a new mechanism for Bolling–Allerod warming. Science 325, 310–314.

Liu, Z., Carlson, A., He, F., Brady, E., Otto-Bliesner, B., Brook, E., Wehrenberg, M., Clark, P., Wu, S., Cheng, J., Zhang, J., Noone, D., Zhu, J., 2012. The Younger Dryer cooling and Arctic climate response to atmospheric CO2. Proc. Natl. Acad. Sci. U.S.A. 109, 11101–11104. http://dx.doi.org/10.1073/pnas.1202183109.

Maher, B., Thompson, R., 2012. Oxygen isotopes from Chinese caves: records not of monsoon rainfall but of circulation regime. J. Quat. Sci. 27, 615–624.

Noone, D., Sturm, C., 2010. Comprehensive dynamical models of global and regional water isotope distributions. In: West, J., Bowen, G., Dawson, T., Tu, K. (Eds.), Isoscapes: Understanding Movement, Patterns, and Process on Earth through Isotope Mapping. Springer, p. 487.

Pausata, F., Battisti, D., Nisancioglu, K., Bitz, C., 2011. Chinese stalagmite δ¹⁸O controlled by changes in the Indian monsoon during a simulated Heinrich event. Nat. Geosci. 4, 474–480.

Schrag, D., Hampt, G., Murray, D., 1996. Pore fluid constraints on the temperature and oxygen isotopic composition of the glacial ocean. Science 272, 1930.

Shakun, J.D., Burn, S.J., Fleitmann, D., Kramers, J., Matter, A., Al-Subbary, A., 2007. A high-resolution, absolute-dated deglacial speleothem record of Indian Ocean climate from Socotra Island, Yemen. Earth Planet. Sci. Lett. 259, 442–456.

Shakun, J., Clark, P., He, F., Liu, Z., Otto-Bliesner, B., Marcott, S., Mix, A., Schmittner, A., Bard, E., 2012. Global warming preceded by increasing CO2 during the last deglaciation. Nature 484, 49–54. http://dx.doi.org/10.1038/nature10915.

Shi, Z., Liu, X., Cheng, X., 2012. Anti-phased response of northern and southern East Asian summer precipitation to ENSO modulation of orbital forcing. Quat. Sci. Rev. 40, 30–38.

Sinha, A., Cannariato, K.G., Stott, L.D., Li, H., You, C.F., Cheng, H., Edwards, L.R., Singh, I.B., 2005. Variability of Southwest Indian summer monsoon precipitation during the Bølling–Allerød. Geology 13, 813–816.

Tan, M., 2013. Circulation effect: response of precipitation δ¹⁸O to the ENSO cycle in monsoon regions of China. Clim. Dyn.. http://dx.doi.org/10.1007/s00382-013-1732-x.

Wang, B., Lin, H., 2002. Rainy season of the Asian–Pacific summer monsoon. J. Clim. 15, 386–396.

Wang, S., Zhao, Z., Chen, Z., Tang, Z., 1987. Drought/flood variations for the last two thousand years in China and comparison with global climatic change. In: Ye, D., Fu, C., Chao, J., Yoshino, M. (Eds.), The Climate of China and Global Climate. China Ocean Press, Springer, Berlin Heidelberg, New York, p. 20 (in Chinese).

Wang, Y.J., Cheng, H., Edwards, R.L., An, Z.S., Wu, J.Y., Shen, C.C., Dorale, J.A., 2001. A high-resolution absolute-dated late Pleistocene monsoon record from Hulu Cave. Science 294, 2345–2348.

Wang, Y.J., Cheng, H., Edwards, R.L., He, Y.Q., Kong, X.G., An, Z.S., Wu, J.Y., Kelly, M.J., Dykoski, C.A., Li, X.D., 2005. The Holocene Asian monsoon: links to solar changes and North Atlantic climate. Science 308, 854–857.

Wang, B., Wu, Z., Li, J., Liu, J., Chang, C.–P., Ding, Y., Wu, G., 2008. How to measure the strength of the east Asian summer monsoon. J. Clim. 21, 4449–4463.

Wang, Y., Liu, X., Herzschuh, U., 2010. Asynchronous evolution of the Indian and east Asian summer monsoon indicated by Holocene moisture patterns in monsoonal central Asia. Earth Sci. Rev. 103, 135–153.

Wu, A., Ni, Y., 1991. The influence of Tibetan Plateau on the interannual variability of Asian monsoon. Adv. Atmos. Sci. 14, 491–504.

Xie, S.P., Hu, K., Hafner, J., Tokinaga, H., Du, Y., Huang, G., Sampe, T., 2009. Indian Ocean capacitor effect on Indo-Western Pacific climate during the summer following El Nino. J. Clim. 22, 730–747.

Yang, Y., Yuan, D.X., Cheng, H., Zhang, M.L., Qin, J.M., Lin, Y.S., Zhu, X.Y., Edwards, R.L., 2010. Precise dating of abrupt shifts in the Asian Monsoon during the last deglaciation based on stalagmite data from Yamen Cave, Guizhou Province, Science China. Earth Sci. 53, 633—641.

Yeager, S., Shields, C., Large, W., Hack, J., 2003. The low-resolution CCSM3. J. Clim. 19, 2545—2566.

Yu, G., Xue, B., Liu, J., 2001. Lake Records from China and the Palaeoclimate Dynamics. China Meteorological Press, Beijing, p. 196.

Yu, G., Xue, B., Li, Y., 2013. Lake level changes in Asia. In: Elias, S.A. (Ed.), The Encyclopedia of Quaternary Science, second ed. Elsevier, Amsterdam, pp. 506—523.

Yuan, D.X., Cheng, H., Edwards, L.R., Dykoski, C.A., Kelly, M.J., Zhang, M., Qing, J., Lin, Y., Wang, Y., Wu, J., Dorale, J., An, Z., Cai, Y., 2004. Timing, duration, and transitions of the last interglacial Asian monsoon. Science 304, 575—578.

Zhang, R., Delworth, T., 2005. Simulated tropical response to a substantial weakening of the Atlantic thermohaline circulation. J. Clim. 18, 1853—1860.

Zheng, W., Wu, B., He, J., Yu, Y., 2012. The East Asian summer monsoon at mid-Holocene: results from PMIP3 simulations. Clim. Past Discuss. 8, 3251—3276.

Zhu, C., Ma, C., Yu, S.-Y., Tang, L., Zhang, W., Lu, X., 2010. A 16000-yr-long pollen record of vegetation and climate changes in Central China. BOREAS 39, 69—76.

---

## Author Comment (AC3) · 21 Jun 2016

Sorry for a typo found in the title: current "P" should be "V" for "Variability".

---

## Author Response (AR1)

General comments: This study looks at how d18Oprecip relates to temperature and precipitation over the East Asian monsoon regions on time scales of variability from seasonal to millennial using an isotope-enabled atmosphere-only climate model. The authors find interesting results that they say mean that d18O speleothem records should be interpreted with caution. I am very supportive of such modelling work to help the palaeoclimate community to better understand how isotopic data might be interpreted. I did though find the paper was quite descriptive and short in its explanations, and would benefit from the addition of further investigation into all the main points they made in order to improve the mechanistic understanding (see comments below). I recommend this be done before the manuscript can be published.

We would like to appreciate the anonymous reviewer for her/his helpful and thoughtful comments. The original comment (Q) and our response (A) are as follows:

Q: Specific corrections: Abstract Line 1: change "Water isotope in precipitation has played a key role" to "Water isotopes in precipitation have played a key role"
A: Done.

Q: Abstract line 1: add references to support the first statement
A: Since it is not allowed to add reference in the abstract part for format reason, the references are included in the first statement of the main text.

Q: Abstract line 5: Although I realize 'thru' is sometimes used for ranges esp. in American English, I would recommend changing "22ka thru 00ka using an isotope-enable AGCM" to "22 ka to 0 ka using an isotope-enabled atmospheric global circulation model (AGCM)".
A: Thanks. It is changed. Also, all the expression of "isotope-enable" in text are changed to "isotope-enabled".

Q: Abstract line 7: "Our study confirms the robustness of the temperature and amount effects on the seasonal cycle over China" – does this statement refer just to the present day? Please add to the text.
A: We have checked the two effects (temperature effect and amount effect) on seasonal cycle timescale in present day and the past slices in the model outputs. The Extended Fig 1 shows an example for North China. We can see the conclusion based on the present day conditions does not change much during the last 22,000 years. But given lack of the observations at seasonal timescale for the past, we would prefer to make the conclusion, in this paper, just for the present day, as compared to the observed monthly data from GNIP. Thanks, we added "in the present climatic conditions" in the text to explicitly clarify this point.

Q: Abstract line 8: "our analysis does not show significant temperature and amount effects over China on millennial and interannual timescales" – do you mean no significant change, or

neither is significantly dominant?

A: The second one. This sentence was changed to "our analysis shows that neither temperature nor amount effect is significantly dominant over China on millennial and interannual timescales".

Q: Introduction page 1 line 13: "Sturm et al., 2010; Noone, 2008" – add an 'e.g.' and perhaps reference to some of the older earlier pioneering papers on this

A: Thanks, it is changed. Also, we added four pioneering papers, such as Dansgaard1964, Grootes1993, Cuffey1995, and Salamatin1998.

Q: Introduction page 1 line 15: ""local temperature effect", whereas the d18O-precipitation relationship in the tropics and low latitudes tends to be associated with the "amount effect" – I would be keen to see a small amount of explanation of these terms for any readers who might be relatively new to the subject.

A: Thanks! We added "a positive correlation between d18O in precipitation and the temperature of ambient air (warmer air provide more energy to rain out 18O-rich water vapor)" to briefly describe the temperature effect, and added "a negative correlation between d18O in precipitation and the accumulated total rainfall at local and upstream regions (stronger tropical precipitation leave less d18O in vapors transported to the subtropics and low latitudes)" to briefly introduce the amount effect in the text.

Q: Page 2 line 9 Change "East Asia locates at the transition zone" to " East Asia is located at"

A: Done.

Q: Page 2 line 10 "still remains as a great controversy" – delete "as"

A: Deleted.

Q: Page 2 line 11 and all other instances of "isotope-enable GCM" change to "isotope enabled GCM"

A: Thanks! We changed all the instances of "isotope-enable" in the text to "isotope-enabled".

Q: Page 2 line 20: delete "proxies".

A: Deleted.

Q: Page 2 line 26: "These experiments are forced by the realistic green house gases (GHGs) concentrations, orbital parameters, land ice sheet and land-ocean mask" – are these all the same boundary conditions as used in the Liu et al. papers, as well as the SSTs/sea-ice?

A: Yes, all the boundary conditions are the same as used in Liu et al. (2009).

Q: Page 2 line 30: "1.6‰ (22ka) to 0.5‰ (0ka)" - please say/reference where you have derived the values from, which have then been linearly interpolated, if I understand right.

A: They came from other scientist's work. References added: Schrag et al. (1996) for 22ka and Hoffmann et al. (1998) for 0ka.

Q: Page 3 line 1: add reference for the GNIP data

A: Reference added: Schotterer and Oldfield (1996).

Q: Page 3 line 3: change "This dataset has sufficient spatial coverage. But majority of: : :" to "This dataset has sufficient spatial coverage but the majority of: : :"

A: Done.

Q: Page 3 line 4: change "there is only12 stations" to "there are only12 stations"

A: Done.

Q: Page 3 line 4: change 'showing' to 'shown'

A: Done.

Q: Page 3 line 18: "For each region, the modeled seasonal cycle are derived from" change to "For each region, the modeled seasonal cycle is derived from"

A: Done.

Q: Figure 2: I know the names of the GNIP stations used are included in the plot, but is it also possible to add in the number that corresponds to the number of the site in figure 1.

A: Sorry I did not understand your question clearly. Do you suggest us to add the station's number into station's name in the text? Or to add a table for 31 GNIP stations? In this revision, we tentatively give the table to you as supportive material. We can put it into the text in the next revision if necessary.

Q: Figure 2: The comparison of the left and right hand graphs is slightly improved as the y-axes have different limits. While I see that this is to maximize the details, it would be easier to make comparisons if the scales were the same. – Actually I realize this is mentioned on page 4 in the penultimate paragraph.

A: Many thanks! The left and right columns share the same y-scale in this revision. The arguments still remain.

Q: Page 3 line 20: why only use the 'GNIP station that has the longest records in that region' in the comparison in figure 2. Have you checked whether there is good correspondence between the one record chosen for each region and the other shorter records in the region? I.e. is each particular record indicative of the overall pattern in the region? Otherwise it seems insufficient reason to choose a particular record based on its length, or say why a longer record is better – e.g. to reduce the impact of interannual variability? Related to this – page 4 paragraph around line 25 – states that the d18O values have somewhat different magnitudes although the phase is a good match with the data, however, there are similar differences in precip and temperature between model and data (as I'd imagine with most models), which might be worth also pointing out in this paragraph.

A: Yes, we did so to reduce the uncertainty of interannual variability. Given the feature of GNIP data's discontinuity (A. many missing within a year; B. observed years is way so short, commonly no more than 10 years, say from 1985 to 1993), as shown in Fig 1(b), the longer

records one station has, the better quality in representing reliable seasonal cycle feature of d18O/temperature/precipitation. The Extended Figure 2 shows an example for NE China: QIQIHAR station, the one used in our study that has longest records among the 4 stations in this region, gives a highly consistent seasonal cycle as the mean. We checked this point for other regions and the same thing happened. The bad continuity and missing of GNIP records in China leads the current method, selecting the site having long and complete observation history, is more important than using as much records as possible.

Q: Page 4 line 8: discusses that the d18O signal from the model in southern China doesn't replicate the seasonal pattern in the data and suggests a resemblance to the 'third mode' as discussed in the following paragraph. However, no mention is made of the fact that the seasonality of precipitation isn't quite right over S China either and how this could influence the mismatch between the model and data d18O.
A: Thanks for this helpful comment. We rewrite the sentence like this: "Instead, the modeled d18O in southern China exhibits a double maximum in spring and fall partly due to the incorrect seasonality of precipitation with its maximum occurred near May-June. The model cannot well reproduce the climatology in this region as it slightly resembles the third mode to be discussed next." Hope it is easier for readers.

Q: Page 4 line 18: change 'implications to the interpretation' to 'implications for the interpretation'.
A: Done.

Q: Page 4 line 20: 'Thus, we would suggest that one should NOT interpret the d18O records around this region simply as the monsoon rainfall amount.' One could also suggest that the boundaries between these different regions could change significantly over time (through glacial-interglacial cycles fro example). It would be useful if the authors could say something regarding this uncertainty and the implications for interpretation of palaeo-isotopic records.
A: Thanks! The changes of Asian monsoon advancing/retreating during glacial-interglacial cycle are much smaller than the amplitude of seasonal cycle of d18O/T/P. Thus, the position of this transition region is robust across glacial-interglacial cycle and change little. Extended Figure 4 shows the model results from time slices other than 00ka (Fig 3a and b). We can see the model suggests a robust and almost stationary "blank region" over the central China.

Q: Page 4 line 31: 'This distinctively different three regions' change to 'These three distinctively different regions'
A: Done.

Q: Page 5, line 10-15 These lines contain a suggestion of why south Asia and East Asia show different correlations between d18O and temp/precip on interannual timescales, but not enough detail to understand the mechanisms for this beyond them having different moisture sources. I suggest a clearer and more detailed explanation is necessary here.
A: Thanks. This point is also closely related to the penultimate question about the relationship between Chinese d18O and Indian monsoon. Please find our response and associated reference

in following part. We added words to discuss the relationship in this revision.

Q: Page 5 line 20: 'using the last 40 years of model output' do you mean where each of the 23 year time slices provides one time point that is the average of the last 40 model years of that simulation. Text could be a bit clearer.
A: Thanks. This sentence is changed to "The millennial climatology is derived from each time slice by averaging the last 30 years out of 50-year raw results." We checked the code, it should be last 30 years rather than 40 years. We revised text and figure captions accordingly.

Q: Page 5 line 25 onwards: it is an interesting result that millennial-scale variability in d18O doesn't reflect high significance in correlation with local temperature or precip. In line with other studies, the authors suggest that d18O over East Asia could be influenced by upstream moisture transport from the Indian monsoon region (similar to Pausata et al). However, they do not investigate this any further in their model so we do not learn as much as we could about what mechanisms are important factors here. The authors have all the data at their disposal and so could look at e.g. correlation at the millennial time-scale of Indian monsoon temp/precip/d18O with d18O over China, and variability in the southerly monsoon winds etc. I would like to see the authors examine what is driving their millennial scale variation further.
A: Thanks! In another paper (Liu and Wen, et al., 2014) with focus on the summer monsoon dynamics, we investigated the variability of d18O on orbital timescale over China, and its relations to Indian precipitation as well as the southerly monsoon winds. Basically, Chinese d18O highly correlated with Indian d18O and precipitation, suggesting a reliable dynamic link between Indian precipitation and Chinese isotope records through amount effect. On the other hand, East Asian monsoon is also influenced by the western North Pacific and South China Sea rather than just the Indian Ocean. The complex circulation determines China's nature having multiple modes of precipitation and d18O. We added some discussion for this part in section 3.2, 3.3, and 4.

Reference: Liu, Z and X Wen et al., 2014: Chinese cave d18O records representing East Asia summer monsoon, Quan. Sci. Rev., 83, 115-128.

Q: Page 5: line 25: Related to the above point, does the seasonality of precipitation/ temperature/d18O change much in the different locations in these 23 time slices? Do the seasonal correlations, interpreted as d18O being affected by the temperature effect in the north and the precipitation effect in the south still hold for the same locations or do the boundaries change from glacial to interglacial time slices?
A: Similarly, please take a look at the Extended Figure 3 for the spatial distribution of temperature/amount effects on seasonal timescale across 20ka, 15ka, 10ka, 5ka, and 0ka. It is shown that the pattern, temperature dominating north and precipitation dominating south, does not change much in the last 20,000 years. You may also find the same clues in Extended Figure 1 for the details of d18O/T/P seasonal cycle over North China, as an example, across the last 22,000 years.

**Extended Figures to Reviewer #1**

ExtFig 1.    The solid seasonal cycle of d18O/temperature/precipitation over North China across the past 22,000 years. The domain is shown in the bottom-right corner.

ExtFig 2.    A demonstration of selecting GNIP site with longest records for Northeast China. The bad continuity and missing of GNIP records in China leads the current method, selecting the site having long and complete observation history, is more important than using as much records as possible.

ExtFig 3.    The spatial patterns of temperature and amount effects at seasonal timescale across the past 22,000 years.

**Table 1.** Basic information of 31 GNIP stations in China

| No. | Station Name | WMO ID | Longitude (°E) | Latitude (°N) | Altitude (m) |
|---|---|---|---|---|---|
| 1 | HONG KONG (KINGS PARK) | 4500400 | 114.2 | 22.3 | 66 |
| 2 | QIQIHAR | 5074500 | 123.9 | 47.4 | 147 |
| 3 | HAERBIN | 5095300 | 126.6 | 45.7 | 172 |
| 4 | HETIAN | 5182800 | 79.9 | 37.1 | 1375 |
| 5 | WULUMUQI | 5182801 | 87.6 | 43.8 | 918 |
| 6 | ZHANGYE | 5265200 | 100.4 | 38.9 | 1483 |
| 7 | LANZHOU | 5288900 | 103.9 | 36.1 | 1517 |
| 8 | YINCHUAN | 5361400 | 106.2 | 38.5 | 1112 |
| 9 | SHIJIAZHUANG | 5369800 | 114.4 | 38.0 | 80 |
| 10 | YANTAI | 5369801 | 121.4 | 37.5 | 47 |
| 11 | TAIYUAN | 5377200 | 112.6 | 37.8 | 778 |
| 12 | CHANGCHUN | 5416101 | 125.2 | 43.9 | 237 |
| 13 | JINZHOU | 5433700 | 121.1 | 41.1 | 66 |
| 14 | TIANJIN | 5452700 | 117.2 | 39.1 | 3 |
| 15 | BAOTOU | 5452701 | 109.9 | 40.7 | 1067 |
| 16 | LHASA | 5559100 | 91.1 | 29.7 | 3649 |
| 17 | CHENGDU | 5629400 | 104.0 | 30.7 | 506 |
| 18 | KUNMING | 5677800 | 102.7 | 25.0 | 1892 |
| 19 | XIAN | 5703600 | 108.9 | 34.3 | 397 |
| 20 | ZHENGZHOU | 5708300 | 113.7 | 34.7 | 110 |
| 21 | WUHAN | 5749400 | 114.1 | 30.6 | 23 |
| 22 | CHANGQING (CUNTAN JIANG) | 5751600 | 106.6 | 29.6 | 192 |
| 23 | CHANGSHA | 5767900 | 113.1 | 28.2 | 37 |
| 24 | ZUNYI | 5771300 | 106.9 | 27.7 | 844 |
| 25 | GUIYANG | 5781600 | 106.7 | 26.6 | 1071 |
| 26 | GUILIN | 5795700 | 110.1 | 25.1 | 170 |
| 27 | NANJING | 5823800 | 118.2 | 32.2 | 26 |
| 28 | FUZHOU | 5884700 | 119.3 | 26.1 | 16 |
| 29 | LIUZHOU | 5904600 | 109.4 | 24.4 | 97 |
| 30 | GUANGZHOU | 5928700 | 113.3 | 23.1 | 7 |
| 31 | HAIKOU | 5975800 | 110.4 | 20.0 | 15 |

[Figure]

[Figure]

**GNIP Observations**

1 HONG KONG (KINGS PARK)
2 QIQIHAR
3 HAERBIN
4 HETIAN
5 WULUMUQI
6 LANZHOU
7 LANZHOU
8 YINCHUAN
9 SHIJIAZHUANG
10 YANTAI
11 TAIYUAN
12 CHANGCHUN
13 JINZHOU
14 TIANJIN
15 BAOTOU

16 LHASA
17 CHENGDU
18 KUNMING
19 XIAN
20 ZHENGZHOU
21 WUHAN
22 CHANGQING (CUNTAN JIANG)
23 CHANGSHA
24 ZUNYI
25 GUIYANG
26 GUILIN
27 NANJING
28 FUZHOU
29 LIUZHOU
30 GUANGZHOU
31 HAIKOU

NE China

**2** QIQIHAR
124°E, 47°N

**3** HAERBIN
127°E, 46°N

**12** CHANGCHUN
125°E, 44°N

**13** JINZHOU
121°E, 41°N

**Mean of 2+3+12+13**

NE China

Mean(2,3,12,13)

[Figure]

[Figure]

**Anonymous Referee #2**

We would like to appreciate the anonymous reviewer for his/her helpful comments! The major revision include performing 4-member AMIP runs and changing the indicator of temperature effect. The original comment (Q) and our detailed response (A) are as follows:

Q: The authors used a list of time slice experiments by an isotope-enabled GCM to evaluate the changes in precipitation d18O on various timescales. It is an interesting work and might give some insights for the interpretation of stalagmite d18O, especially for the paleoclimate reconstructions in Asia. I do not know whether these experiments are the same as those in Liu et al. 2014 QSR or not. The authors should clarify this in the section of model description. These experiments are no doubt useful for exploring the interpretation of the precipitation d18O over the East Asian on different time scale. However, I am afraid that the present experiment design is not reasonable enough for examining the changes in d18O, especially on the seasonal and interannual timescales. The present 0Ka experiment may neglect some major changes in boundary conditions and can not directly compare to the modern GNIP observations. Are the greenhouse gases and sea surface temperature kept constant? Why do the authors not employ the observed GHG and SST to force the atmosphere model? This experiment is necessary and do not need much time. I strongly recommend to add this experiment and to reanalyze the results.

A: Yes, the numerical experiments are the same as those in Liu and Wen, et al. (QSR, 2014) but with different research goals. We added sentences in section 2 to clarify this point. In Liu's paper (2014), we discussed the dynamic linkage between Chinese d18O and East Asian summer monsoon, whereas in this paper we would like to discuss the robustness of interpreting d18O records in terms of two effects on three different timescales: seasonal, interannual, and millennial.
At the early stage of this work, we planned to focus on 4 timescales: millennial, interdecadal, interannual, and seasonal. But the big problem is lack of observed d18O record on interannual-to-interdecadal timescales. Most of the records from GNIP network have no more than 8 years (1985-1993), as shown in Fig 1b. Thus, we removed the "interdecadal" and focus on the remaining three, among which the "interannual timescale", relatively, could be the one lacking observations most. In general, the interannual variability of d18O or other variables include two sources: climate system internal variance and responses to external forcing. The observed d18O records are too short to reliably account for both. Our 00ka slice was driven by 1950 boundary conditions and was integrated for 50 years, which is able to provide enough interannual information than GNIP for internal variability problem, but for forcing-response problem. This is the shortage of current experiments. Appreciate the reviewer's kind suggestion! In this revision, we performed a 4-member ensemble AMIP-type runs covering the period 1975-2004 (30 years in total) with external forcing of observed SST/SICE and GHGs. As an example, Extended Figure 1 shows the interannual variabilities of JJA d18O in the new runs and GNIP at Hong-Kong, a site having longest records in China (as shown in

Fig. 1b). We found that: 1) the results at interannual scale change evidently. So we revised Figure 3 and 4 and related text accordingly. 2) almost no change can be found at seasonal scale, since the amplitude of seasonal cycle is much greater than the interannual variance (also can be found in Fig. 4).

Moreover, we keep both the interannual results from 00ka and new AMIP runs in Fig. 4 to help readers to understand their differences. Apparently they still fall into the same "interannual regime" with little differences. It could be noted that not easy to separate internal variance (00ka results) and forcing-response variance (newly conducted AMIP runs) at this timescale.

Q: The authors use a series of time-slice experiments for the last 22 ka to evaluate the "temperature effect" and "amount effect" on millennial time scale in different regions in East Asia. I think that the author should present the long-term changes of precipitation d18O in these model simulations and compare them with the proxy records. If the outputs of these experiments capture the variations in the proxy time series, then it's robust to test the interpretation of the precipitation d18O on millennial time scale by using the model simulation. Otherwise, the bias in the model itself will mask the real processes which affect the precipitation d18O changes. This is fundamental to the model simulation. The authors must cross check the model outputs with the real observations and then come to the conclusion.

A: We compared the model results (d18O, precipitation, and meridional winds) with proxy data in Liu and Wen, et al. (QSR, 2014). It is shown that the model successfully reproduces the observed orbital and millennial variability as compared to multiply proxies and generates reliable monsoon-associated anomalous circulation (in Liu et al., QSR, 2014, Fig 2e shows model results by comparing with d18O proxy and V winds; Fig 2f and 2g shows the comparison of modeled precipitation and other lake sediment proxies). This forms the solid base for present investigation.

Reference: Liu, Z and X Wen et al., 2014: Chinese cave d18O records representing East Asia summer monsoon, Quan. Sci. Rev., 83, 115-128.

Q: As shown in figure 3, the authors correlate the annual mean d18O weighted with precipitation to the DJF temperature and JJA precipitation on the interannual and millennial time scales (panel c-f) and then use this statistic result to argue the "amount effect" and "temperature effect". This is totally wrong! Because the annual mean temperature may not change the same way as the DJF temperature, and also the varied precipitation seasonality (as shown in figure 2) in different regions may deny the dominant contribution of summer precipitation to the annual precipitation.

A: Many thanks! The observed d18O records in speleothem, fundamentally, reflects precipitation-weighted annual mean. For Asian monsoon region, it could be considered that

these cave d18O records mostly reflect summertime rather than wintertime information. Appreciate the reviewer's suggestion. Here, we re-examine this problem for millennial (Extended Figure 1) timescale, as an example. It is shown that the p-weighted annual mean of temperature and JJA mean precipitation could be the appropriate ones accounting for temperature effect and amount effect. They are even more reasonable than monthly equal-weighted annual mean by emphasizing rain-season footprint. We revised Figure 3 and text accordingly in this revision.

We further investigate the robustness of interannual patter (as above) of temperature/amount effects across the past 22,000 years (say, 20ka, 15ka, 10ka, 5ka, and 0ka), as shown in the Extended Figure 3. It is shown that the weak correlation region (the blank region) does not change much, suggesting the conclusion that one should be very cautious in interpreting d18O records for this area on interannual timescale still remains.

Q: Page 1 line 19, the citation of Yuan et al., 2004 is wrong. It presents the speleothem d18O record from southern China.

A: Thanks, we moved this item to speleothem part.

**Extended Figures to Reviewer #1**

ExtFig1.    Timeseries of JJA mean d18O in 4 AMIP runs and GNIP observation at Hong-Kong station.

ExtFig2.    Check the indicators for temperature and precipitation effects. We select $CC(d18O_{p-wgt}, TS_{p-wgt})$ as the temperature effect indicator; $CC(d18O\_p-wgt, PREC_{JJA})$ as the amount effect indicator.

ExtFig3.    The solid spatial pattern of temperature and amount effects at interannual timescale across the past 22,000 years.

[Figure]

**Millennial Timescale**

[Figure]

**Interannual Timescale**